# Attention Mechanisms Perspective: Exploring LLM Processing of Graph-Structured Data

**Zhong Guan** [* 1 2]  **Likang Wu** [* 1 2 3]  **Hongke Zhao** [1 2 3]  **Ming He** [4]  **Jianping Fan** [4]

## Abstract

Attention mechanisms are critical to the success of large language models (LLMs), driving significant advancements in multiple fields. However, for graph-structured data, which requires emphasis on topological connections, they fall short compared to message-passing mechanisms on fixed links, such as those employed by Graph Neural Networks (GNNs). This raises a question: "Does attention fail for graphs in natural language settings?" Motivated by these observations, we embarked on an empirical study from the perspective of attention mechanisms to explore how LLMs process graph-structured data. The goal is to gain deeper insights into the attention behavior of LLMs over graph structures. We uncovered unique phenomena regarding how LLMs apply attention to graph-structured data and analyzed these findings to improve the modeling of such data by LLMs. The primary findings of our research are: 1) While LLMs can recognize graph data and capture text-node interactions, they struggle to model inter-node relationships within graph structures due to inherent architectural constraints. 2) The attention distribution of LLMs across graph nodes does not align with ideal structural patterns, indicating a failure to adapt to graph topology nuances. 3) Neither fully connected attention nor fixed connectivity is optimal; each has specific limitations in its application scenarios. Instead, intermediate-state attention windows improve LLM training performance and seamlessly transition to fully connected windows during inference. Source code: LLM4Exploration

[*]Equal contribution [1] College of Management and Economics, Tianjin University, Tianjin, China [2] Laboratory of Computation and Analytics of Complex Management Systems ,Tianjin University,Tianjin, China [3]ai-deepcube [4]AI Lab at Lenovo Research, Beijing, China. Correspondence to: Hongke Zhao <hongke@tju.edu.cn>.

*Proceedings of the 42nd International Conference on Machine Learning*, Vancouver, Canada. PMLR 267, 2025. Copyright 2025 by the author(s).

## 1. Introduction

LLMs have garnered significant attention, achieving remarkable success in language processing and demonstrating effective transferability to various other domains (Goyal et al., 2024; Ouyang et al., 2022; Pi et al., 2024; Singh et al., 2023; Wu et al., 2024; Zhao et al., 2024; Wu et al., 2023). This trend has inspired the graph machine learning community to delve into the application of LLMs within their field (He et al., 2024; Chen et al., 2024b; Huang et al., 2024; Kong et al., 2024; He & Hooi, 2024; Liu et al., 2025; Guan et al., 2024). However, recent studies reveal that existing LLMs for graphs fail to deliver satisfactory performance on graph-structured data, pointing to undiscovered challenges that hinder the deployment of LLMs in this context (Luo et al., 2024).

Attention mechanisms are a critical component of LLMs success, effectively linking tokens to enable models to comprehend complex contexts and domain-specific knowledge (Xiao et al., 2024; Hsieh et al., 2024; Yu et al., 2024). Despite the vast potential of attention mechanisms, research into their application on graph-structured data remains largely unexplored, lacking systematic analysis. Therefore, we embarked on an investigation from the perspective of attention analysis, aiming to uncover unique phenomena of LLMs attention on graph data structures and to validate our hypotheses regarding LLMs' behavior on such data. Our work seeks to fill this research gap and establish a clear direction for future studies in the field.

In this paper, we conducted our study based on the following hypotheses, and uncovered new issues and phenomena.

**Q1: Do the attention distribution of LLMs change before and after training with finetuning ? Can LLMs correctly utilize graph structures?**

We first compared the distribution curves of attention scores for node tokens and text tokens before and after LLMs training. We then conducted hypothesis testing to clarify the following points: whether the attention on node tokens has shifted; whether the attention on text tokens has shifted; and whether the attention distributions between node tokens and text tokens are consistent.

The results showed that after training, the LLMs' attention towards node tokens indeed underwent a significant shift. This suggests that the LLMs have developed an initial capability to recognize graph-structured data. Simultaneously, our hypothesis testing revealed that the attention distribution of LLMs within nodes exhibits extreme tendency. However, in subsequent experiments where we disrupted the connectivity information, we found that even when the topological connection information was randomly shuffled, it had almost no effect on the LLMs' performance. This indicates that the LLMs did not effectively utilize the correct connectivity information.

**Q2: Can LLMs allocate attention to different types of graph nodes in a manner consistent with the structural properties of the graph?**

When processing graph data inputs, LLMs calculate attention scores between node tokens to weigh the importance of different nodes relative to each other. Through our visualization experiments, we found that the attention scores between different node tokens in LLMs do not adequately match the graph structure. Specifically, under sequential conditions, the attention distribution of node tokens exhibits a U-shaped or long tail, which deviated from our idealized assumptions. Ideally, the model should focus more on central nodes and allocate attention in a hierarchical, diminishing manner.

Meanwhile, our analysis revealed that the attention paid by text tokens to node tokens more closely matches our ideal expectations. This indicates that the current limitation in LLMs lies not in the interaction between text and nodes but in the modeling of connections between nodes.

**Q3: Which is more suitable for LLM's graph-structured tasks: the fully connected perspective of LLMs or the fixed-link perspective of GNNs?**

We introduce a specific metric, the Global Linkage Horizon (GLH), to measure the visibility range between nodes in LLMs. Through extensive experiments adjusting the GLH, we found that neither the fully connected view of LLMs nor the fixed-linkage view of GNNs represented the optimal attention perspective for LLMs.

Intermediate perspectives that incorporate certain topological link information achieve superior performance during training and yield unexpected improvements when used solely for inference. Specifically, models trained with a smaller linkage horizon can be effectively deployed with a larger linkage horizon. This transferability from small to large perspectives addresses practical deployment challenges while enhancing model performance.

In this work, our primary objective is to identify unique phenomena and explore the causes and potential impacts of these phenomena, aiming to provide new perspectives on

how LLMs process graph-structured data, thereby guiding the direction of future discoveries for the community.

Our contributions in this work are summarized as follows:

- We conducted a visualization and analysis of LLM attention on graph-structured data. To our knowledge, this is the first empirical study to investigate LLMs for graph machine learning from the perspective of attention.

- Our analysis reveals that LLMs fail to effectively leverage the connectivity information in graphs. We delve into this issue by examining two major aspects: the distribution of attention scores and the scope of attention windows.

- We identify "Attention Sink" issues similar to those observed in natural language tasks, as well as a unique phenomenon we term "Skewed Line Sink" specific to graph data. Drawing on the experience of the NLP community, we can explore methods to correct these biases to improve model performance.

- Through a series of experiments, we identified multiple phenomena and current challenges faced by LLMs in graph machine learning applications. Our work provides valuable insights and guides future research efforts in this field.

## 2. Related Work

### 2.1. Analysis of Attention in LLMs

With the remarkable attention that LLMs have garnered across various communities, pioneering studies have begun to focus on the attention mechanisms within LLMs and their distribution (Ruan & Zhang, 2024; Acharya et al., 2024; Makkuva et al., 2024; Liang et al., 2024). Stream-LLM (Xiao et al., 2024) was among the first to identify the phenomenon of "Attention sinks", where semantically limited initial tokens can receive disproportionately higher attention scores. The study suggested maintaining these special tokens during long-text inference for better performance. Building on StreamLLM, Yu et al. (2024) found that correcting partial attention sinks can yield performance improvements without additional training. Duan et al. (2024) explored the relationship between attention and sentence uncertainty, while Hsieh et al. (2024) investigated the "lost-in-the-middle" phenomenon in RAG, focusing on the interaction between retrieved document ranking and attention.

### 2.2. Attention Window

We define the attention window as the visibility range between tokens within a single layer of a neural network. In models like BERT (Kenton & Toutanova, 2019), the attention view is bidirectionally fully connected, allowing each

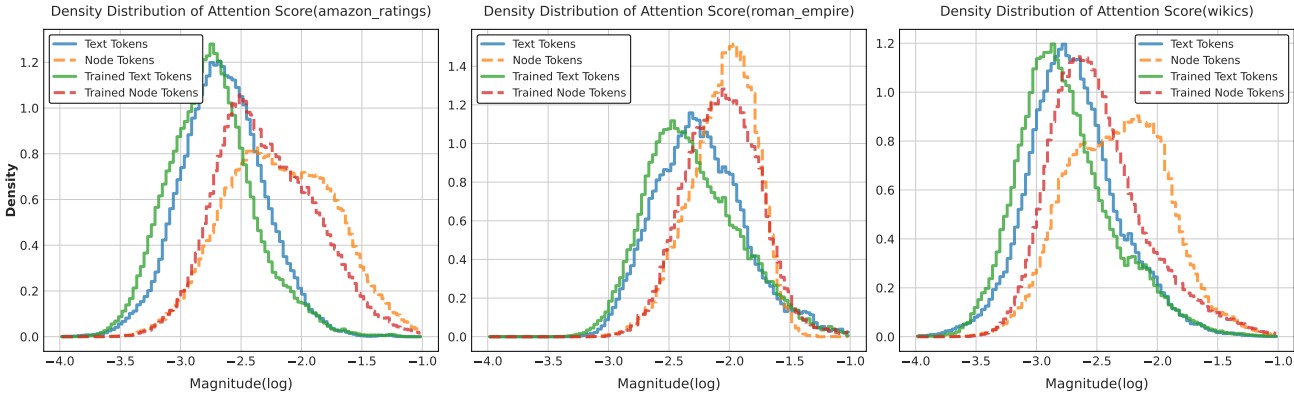

*Figure 1.* Attention distribution of different types of tokens before and after training. With Amazon-Ratings on the left, Roman-Empire in the middle, and Wikics on the right. The attention values have undergone log scaling and are plotted as a density distribution Figure.

token to attend to all other tokens. Conversely, in GPT series models (Brown et al., 2020), a unidirectional causal mask is used, restricting each token to only see preceding tokens, resulting in a lower triangular attention matrix. Similarly, in GNNs, the visibility is defined by fixed connections, where each node can only see its directly linked neighbors (Kipf & Welling, 2022). In the context of applying LLMs to graph data mining, some works have simply adapted LLMs to use either fixed-link or bidirectionally fully connected attention window (Yang et al., 2024; Zhu et al., 2024b). However, these adaptations lack thorough experimental analysis and deeper exploration of the attention window's impact on model performance.

## 3. Empirical Study

Our experiment and analysis summary are based on multiple datasets. In addition to the presentation part, some detailed settings and results can be specifically seen in **Appendix**.

### A1. Changes in Attention Distribution

Fine-tuned LLMs(LLaGA (Chen et al., 2024b)) show relative improvements on graph tasks. To investigate whether LLMs can recognize the differences between graph-structured data and natural language data, as well as effectively utilize graph structural information (Q1), we compare the changes in token attention focus before and after training.

**Setting.** $H_0$: The attention focused on node tokens does not change before and after fine-tuning. $H_1$: The attention focused on text tokens changes before and after fine-tuning. $H_2$: The attention distributions for node tokens and text tokens are not consistent. It is worth noting that we follow the recognized LLaGa guidelines. For more detailed information, see Appendix J.

We compared the attention score distributions for node tokens and text tokens before and after training in Figure 1.

*Table 1.* Statistical Analysis of Attention Score Distributions(Roman-Empire Dataset). *Note: ** indicates p-value <0.01, otherwise p >0.05 for t-test and KS test. JS Divergence values are provided as this.*

| Comparison | T-Test | KS Test | JS |
|---|---|---|---|
| Before vs After (Text) | 44.061** | 0.092** | 0.0064 |
| Before vs After (Node) | 0.461 | 0.049** | 0.0099 |
| Node vs Text (Before) | -60.455** | 0.279** | 0.0753 |
| Node vs Text (After) | -78.651** | 0.316** | 0.0791 |

Our analysis reveals a marked shift in the attention distribution for node tokens, demonstrating that LLMs begin to recognize node tokens. In a nutshell, it is observed that LLMs tend to align the attention distribution of node tokens and text tokens.

To statistically validate these observations, we performed T-tests, Kolmogorov-Smirnov (KS), and Jensen-Shannon Divergence (JS) on the attention score distributions, with the results summarized in Table 1.

Our statistical analysis reveals a distinctive outcome when comparing the attention score distributions before and after training, specifically for node tokens. The KS test indicates a significant change in the distribution of attention scores post-training, while the t-test suggests that the mean attention scores remain unchanged. Given that the KS test focuses on the overall distribution and the t-test emphasizes the mean value, this discrepancy highlights the nuanced changes in attention patterns.

Additionally, the JS distance further supports these findings by showing that the overall distribution of attention scores for nodes changes more significantly compared to text tokens. This implies that while the average attention levels remain consistent, the distribution of attention scores exhibits a bimodal trend, indicating that LLMs develop distinct attention patterns for certain nodes, leading to a

polarization in attention allocation.

## A2. Structure Information Disruption Experiment

After discovering that LLMs can perceive graph-structured data, we further investigated whether they can effectively utilize this type of data. To explore this, we designed disruption experiments with varying levels of connectivity information alteration and observed changes in model performance. Ideally, as the level of disruption increases, the model's performance should deteriorate. Additionally, we collected attention scores from the model to provide new insights into its behavior in graph data.

**Setting.** We divided the disruption experiments into four levels: (I) Swapping entire sets of child nodes between pairs of first-order nodes. (II) Exchanging random numbers of child nodes between first-order nodes, and further disrupting the information structure. (III) Randomly shuffling the positions of first-order and second-order nodes, even allowing original second-order nodes to become first-order nodes. (IV) Incorporating unrelated nodes and performing random substitutions. And (Raw) keeping information structure.

The results of these experiments are shown in Table 2. Disappointingly, the LLMs showed no significant reaction to the perturbation of graph-structured connectivity information across half of the datasets. Specifically, in the majority of cases(Wikics, Pubmed), there was no significant change in model performance under disruption levels (I) and (II). Only at higher levels of disruption (III) or (IV) did we observe a decline in performance. However, GNNs that passed the WL-test have been continuously decreasing.

In contrast, only the Roman dataset presented an ideal scenario where the model's performance degraded progressively with increasing levels of disruption. This behavior aligns with our hypothesis that the model effectively leverages graph-structured information. The consistent degradation of performance on the Roman dataset suggests that LLMs can indeed exploit structural information when it is sufficiently represented and not overly disrupted. In other datasets, the model failed to demonstrate such a clear pattern, indicating limitations in its ability to utilize graph structure.

To gain deeper insights, we meticulously examined changes in the distribution of attention scores assigned by neighboring nodes to central nodes before and after training, under varying perturbations. The results are illustrated in Figure 2. Figure 2 delineates the distribution of attention pre- and post-training, revealing a diminished focus on graph structure through reduced attention from neighbor to central nodes in datasets where graph structure information was not effectively utilized. Conversely, on the Roman-Empire dataset, there is an observed increase in such attention, signifying the model's learned utilization of graph structure information.

**Overall(Q1)**, while LLMs exhibit an awareness of graph-structured data, their current mechanisms limit their ability to effectively utilize this information.

In the subsequent subsection, we will delve deeper into the distribution of attention scores to further explore how LLMs process graph-structured data.

## B. Adaptability of Attention Distribution to Graph Data

When the graph structure is input to the LLMs in natural language form, the model assigns different attention scores to each graph node. By analyzing the attention scores of different types of nodes, we aim to examine whether the LLM can effectively adapt to the graph data structure (Q2) and assess whether its attention distribution aligns with the ideal state—that is, whether it can reasonably allocate attention according to the topological structure of the graph.

**Setting.** We emulate the most common instruction construction methods (such as InstrutGLM (Ye et al., 2023) and LLaGA (Chen et al., 2024b)) to describe graph link structures in natural language form shown in Figure 4. During this process, the number of neighbors for each central node varies, meaning that fixed positions in the input sequence may be occupied by different types of nodes or text tokens. To eliminate interference from factors such as sequence length and position, which could affect the attention scores assigned to node tokens, we introduce two operations: padding and random shuffling. These operations ensure that the input instructions and nodes have a fixed length and position, as illustrated in Figure 4.

### B1. Position Bias Interferes with Attention Adaptation

Through our experiments, we recorded attention scores for different nodes at fixed positions. As shown in Figure 3 Upper, node tokens exhibit a slash trend or U-shaped distribution of attention towards node tokens, with attention scores sharply increasing for the final nodes. This distribution aligns with the common attention distribution curve observed in language models. Additionally, we attribute the lower attention scores for the initial nodes to their not being positioned at the forefront of the entire text.

However, it reveals an issue: under the graph data structure, the LLM's attention to important nodes does not adequately adapt to the graph structure. Specifically, in the ideal state of graph machine learning, the model should prioritize central nodes and gradually decrease attention to neighboring nodes in hierarchical order. Alternatively, if there are super-nodes, a random arrangement should yield an average trend. Instead, the curve we obtained is entirely different from the ideal state, with the importance of central nodes significantly lagging behind other nodes.

In our exploration of this issue, we conducted a more detailed analysis of the attention patterns between text tokens

*Table 2.* Connectivity Information Disruption Performance. Node classification results (accuracy(%) ±std) for 4 runs on four real-world datasets and five levels of disruption. The model used is LLama2-7B, with node sampling configured as 8x8.

| Dataset | Raw | (I) | (II) | (III) | (IV) |
|---|---|---|---|---|---|
| Wikics | 0.7862± 0.007 | 0.7847± 0.004 | 0.7907± 0.0035 | 0.7670± 0.003 | 0.7080± 0.005 |
| Pubmed | 0.8367±0.003 | 0.8363±0.001 | 0.8364±0.002 | 0.7698±0.003 | 0.7835±0.001 |
| Amazon-Ratings | 0.4486±0.002 | 0.3980±0.003 | 0.3977±0.002 | 0.3975±0.003 | 0.3913±0.001 |
| Roman-Empire | 0.8089±0.001 | 0.7918±0.001 | 0.7910±0.002 | 0.6290±0.002 | 0.5784±0.002 |

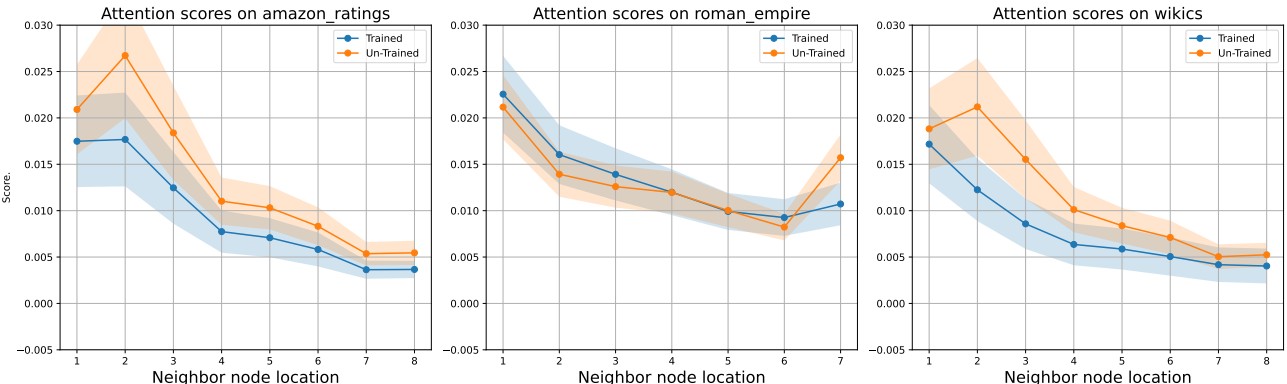

*Figure 2.* The attention scores from neighboring nodes to the central node, both before and after training, were presented as mean values with standard deviations, using a 1:8 sampling ratio.

and node tokens in Figure 3(Lower). The attention paid by text tokens to node tokens aligns more closely with the ideal scenario: first-order nodes receive higher attention scores compared to their surrounding second-order nodes. The attention scores exhibit a fluctuating upward trend as a function of node position, with peaks occurring at positions corresponding to structurally significant nodes.

This indicates that the ability of LLMs to capture relationships between text and nodes is already quite robust; the aspect that requires further development is the connectivity among nodes within LLMs.

When exploring whether there are inconsistencies in the attention paid by node tokens and text tokens to text tokens, the results showed no significant differences between the two. The locations of attention sinks or minor fluctuations were largely consistent, indicating a high degree of similarity. This suggests that both node tokens and text tokens exhibit consistent attention patterns towards text tokens.

**Remark(Q2).** We found that the attention mechanism does not adapt well to sequentially input graph-structured data, as its distribution does not meet the ideal scenario. Unlike NLP tasks, where the mismatch distribution is acceptable due to uncertain positions of important text, graph-structured data contains prior importance information. As a result, LLMs with this issue cannot match the performance of GNNs that focus on graph structure. Furthermore, some studies

on RAG indicate that the placement of tokens in different positions significantly affects attention (Hsieh et al., 2024).

**B2. Nodes Interaction Attention Score**

To delve deeper into the nuances of attention distribution among node tokens, we conducted a heatmap visualization analysis of the attention score matrix, uncovering several novel phenomena.

More precisely, Figure 5 shows the attention interaction matrix across all nodes. Based on this, we observed the so-called "Attention sink" phenomenon (Xiao et al., 2024), which manifests in two distinct patterns across most graph datasets. The first pattern is a simple "Attention sink" as shown in Figure 5(Left), where certain positions consistently attract higher attention scores without significant topological or sequential information.

The second pattern exhibits a unique "diagonal" characteristic specific to graph data. Typically, diagonals near the main diagonal exhibit higher attention values due to nodes paying more attention to their neighbors; however, as depicted in Figures 5(Right), there exists a diagonal with notably higher attention scores compared to surrounding diagonals. This pattern was not found in textual inspections and suggests that the model may be learning to capture unexpected spatial patterns or path dependencies, possibly arising from the inherent properties of the graph structure. Given its distinction from simple adjacency relationships, we term this pattern

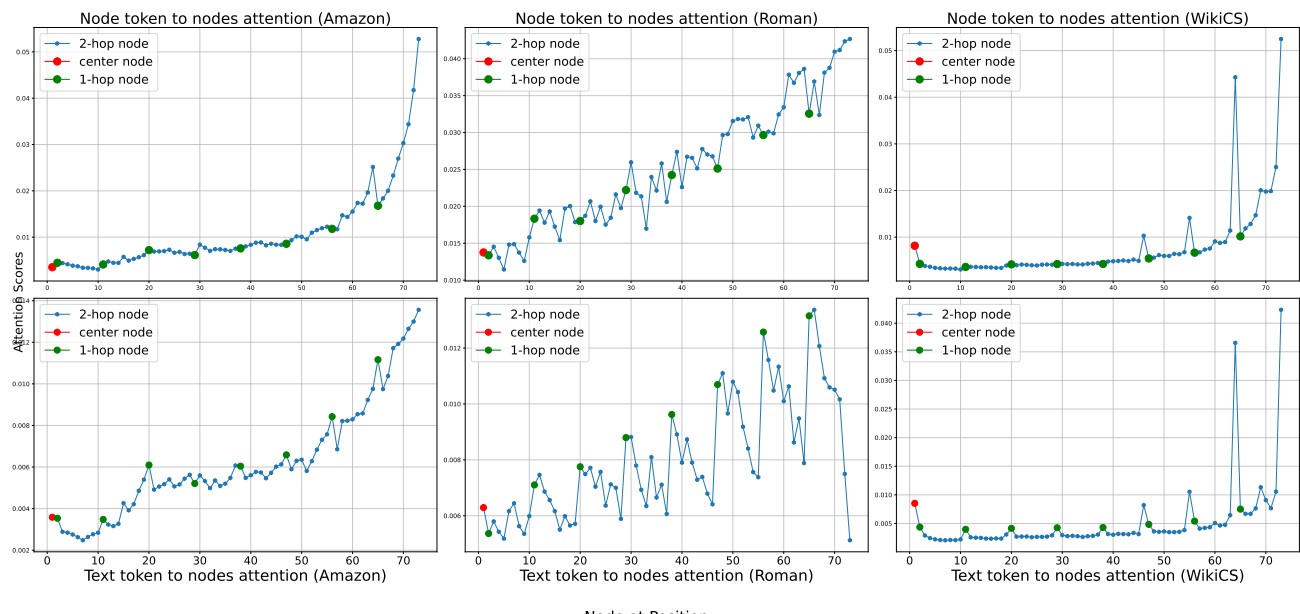

*Figure 3.* Illustration of all tokens to nodes attention. The x-coordinate refers to the relative position of the node token in the entire node list. We collected the attention scores of all tokens towards node tokens and plotted them in a line graph according to the relative position of nodes within the instructions. **Upper:** the mean values of attention scores from nodes(Querys) to nodes(Keys). **Lower:** the mean values of attention scores from texts(Querys) to nodes(Keys). The plot annotates nodes at different hierarchical levels. Our node sampling is 8*8.

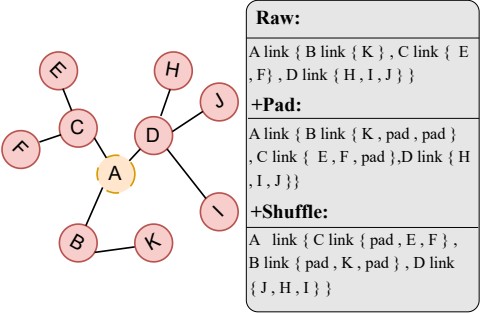

**Raw:**

A link { B link { K } , C link { E , F} , D link { H , I , J } }

**+Pad:**

A link { B link { K , pad , pad } , C link { E , F , pad },D link { H , I , J }}

**+Shuffle:**

A link { C link { pad , E , F } , B link { pad , K , pad } , D link { J , H , I } }

*Figure 4.* Padding and Random Shuffling to ensure fixed sequence length and position. This minimizes interference with attention scores for other reasons.

"Skewed Line Sink."

The emergence of "Attention sink" and "Skewed Line Sink" phenomena interferes with the proper allocation of attention between nodes in LLMs, preventing them from effectively utilizing graph structural information. However, based on these findings, we can engage more effectively with the NLP community to correct these biases.

## C. Attention Window for Graph-Structured Data

In recent works (Kim et al., 2022; Joshi, 2020), it has been shown that transformers can be viewed as fully connected graph-attention models, while contemporary decoder-only LLMs function as unidirectional fully connected Graph Transformer (GT) models. For given graph-structured data, the attention window of LLMs manifests as a lower triangular matrix representing full connectivity, whereas the attention window of GNNs is a fixed-linkage adjacency matrix. These two extremes represent the spectrum of attention windows in graph-structured data. The attention window determines how the model captures relationships between nodes and is a critical factor influencing the effectiveness of graph structure learning. In this section, we explore how LLMs learn and utilize graph structures from different visible perspectives, seeking the optimal linkage perspective between fully connected and fixed-linkage views(Q3).

**Setting.** We introduce a **global linkage horizon** $k$ to indicate the visibility range of node tokens under the attention window. $k$ ranges from 0 to 2L, where $L$ represents the number of hops or neighborhood radius from the central node. As shown in Figure 6, when sampling subgraphs with $L = 2$ hops: $k = 0$ means all nodes can only see themselves, $k = 1$ indicates all nodes can see their first-order neighbors, and $k = 4$ means nodes can see all other nodes. Among these, $k = 0$ and $k = 4$ represent Attention Windows without effective link information, while $k = 1, 2, 3$ include partial graph link information. Additionally, considering that

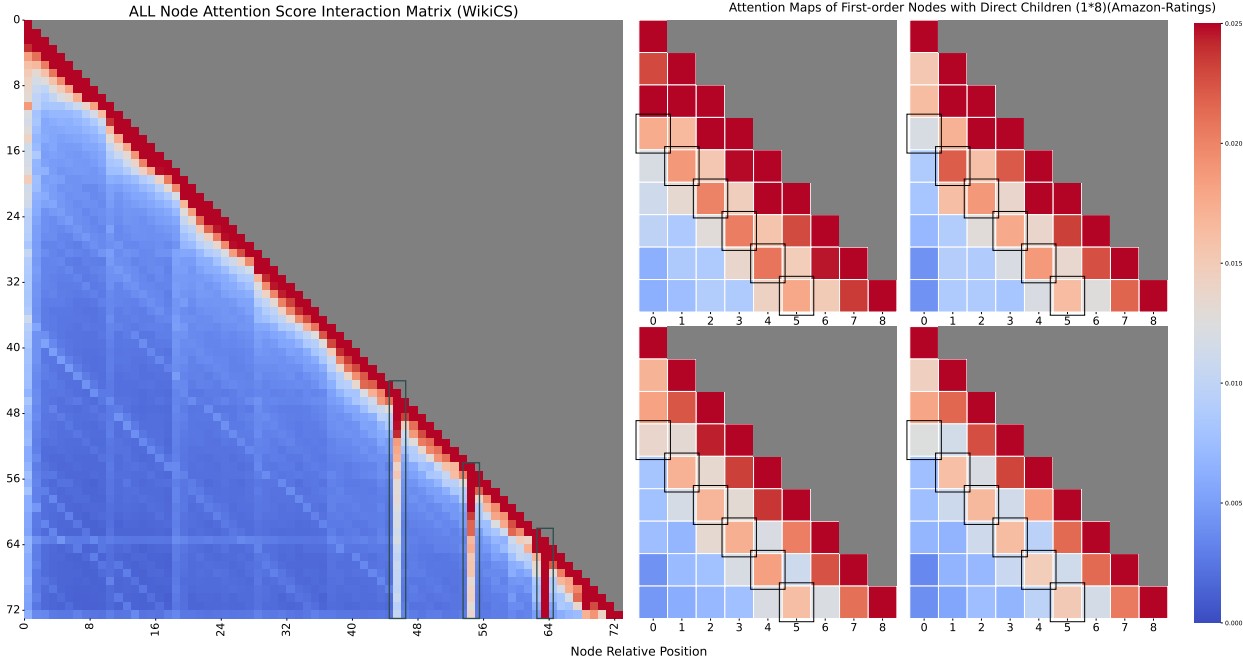

*Figure 5.* Illustration of attention score interaction matrix(Nodes). **Left:** The left panel displays the attention interaction matrix between all nodes (1 central node + 8 first-order nodes + 8*8 second-order nodes), averaged across all heads and layers. **Right:** The right panel shows the attention interaction matrices between first-order and second-order nodes, with four selected groups highlighted for detailed analysis. We highlight the "attention sink" with a gray box and the "Skewed Line Sink" with a black box.

*Table 3.* Model performance(accuracy(%) ±std) for 4 runs on four real-world datasets and under different values of k

| k | Wikics | Roman | Amazon | Pubmed |
|---|--------|-------|--------|--------|
| $k^4_{\text{unidi}}$ | 77.49±0.21 | 80.73±0.0 | 45.19± 0.2 | 82.92±0.1 |
| $k^4_{\text{bidi}}$ | 67.81±0.06 | 81.38±0.0 | 43.79± 0.4 | 82.47±0.4 |
| $k^3_{\text{unidi}}$ | 77.96±0.13 | 81.68±0.02 | 45.46± 0.0 | **83.36±0.1** |
| $k^3_{\text{bidi}}$ | 74.93±0.26 | 81.61±0.0 | 45.63± 0.1 | 81.98±0.1 |
| $k^2_{\text{unidi}}$ | **78.90±0.17** | 82.87±0.02 | 44.67± 0.0 | 82.89±0.3 |
| $k^2_{\text{bidi}}$ | 77.42±0.19 | 82.05±0.02 | 45.18± 0.2 | 82.67±0.1 |
| $k^1_{\text{unidi}}$ | 78.15± 0.02 | **83.12±0.0** | 45.95± 0.1 | 80.49±0.2 |
| $k^1_{\text{bidi}}$ | 78.43± 0.09 | 83.05±0.01 | **46.17± 0.0** | 79.81±0.2 |

LLMs are mostly unidirectional decoders, while GNN links are bidirectional, we incorporate both unidirectional ($k_{\text{unidi}}$) and bidirectional ($k_{\text{bidi}}$) links into our empirical study.

**C1. Optimal Visible Perspective**

We conducted multiple experiments with different $k$ values, and the results are summarized in Table 3. From the table, it is evident that when the training and inference $k$ values remain unchanged, the fully connected view ($k = 4$) of LLMs cannot achieve the best performance. In contrast, intermediate views ($k = 2, 3$), which contain certain topological link information, achieve better results. The fixed-linkage view

($k = 1$) is also considered to perform well.

However, settings with $k \neq 4$ require pre-labeling node tokens and modifying the attention window, posing deployment challenges in **real-world scenarios**. Therefore, we continue our exploration.

**C2. Unidirectional vs. Bidirectional Attention**

To further investigate the impact of unidirectional and bidirectional attention, we conducted comparative experiments in Table 4. The mutual transfer between unidirectional and bidirectional attention is difficult to achieve better results, and the loss caused by transfer increases as the value of k grows.

**C3. Transferability across Different Visible Perspectives**

We also tested the transfer ability of models trained at one $k$ value and inferred at others in Table 4.

Small-to-large: Surprisingly, switching from smaller to larger $k$ values does not weaken model performance but rather enhances it. Specifically, we can achieve better performance at inference time with k=4 by training the model at $k = 2$ or $k = 3$, compared to training directly at $k = 4$. Since $k = 4$ is the fully connected view of LLMs, no modification is required during real-world inference, addressing the deployment difficulties mentioned in subsection C1.

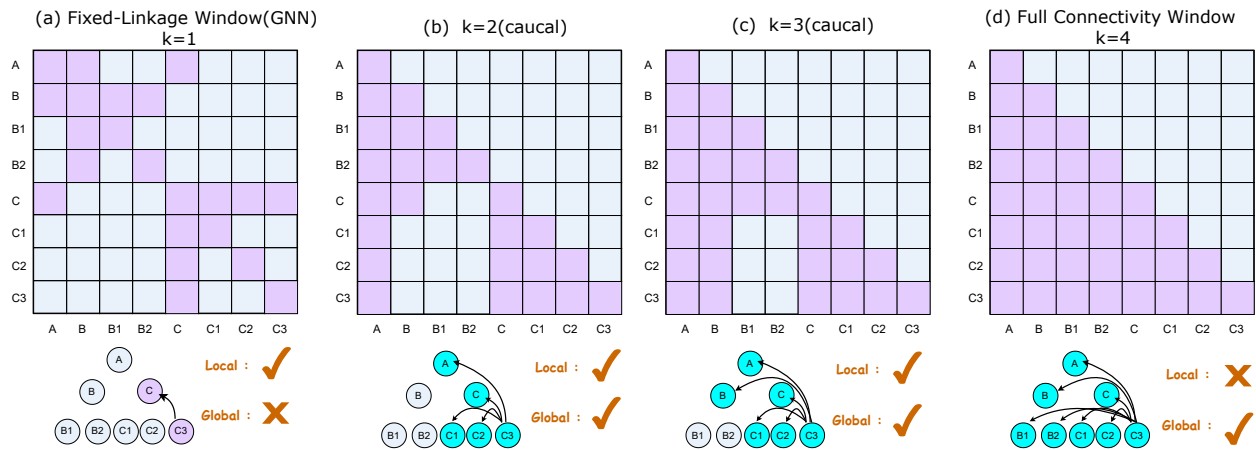

*Figure 6.* Illustration of different global linkage horizon k. From left to right, the images represent k=1 to 4. To demonstrate the field of view of GNNs, the image for k=1 is depicted bidirectionally, while the rest are shown unidirectionally.

*Table 4.* Transfer Performance of Models across Different $k$ Values. **Rows** indicate the $k$ value used during training, **Columns** represent the $k$ value utilized during testing. The best performance for each $k$ value transfer is highlighted in **gray**, with the overall best performance **bolded**. The best result of LLM testing under normal window ($k_{\text{unidi}}^4$) conditions is indicated with an underline.

| Test \ Train | $k_{\text{bidi}}^4$ | $k_{\text{unidi}}^4$ | $k_{\text{bidi}}^3$ | $k_{\text{unidi}}^3$ | $k_{\text{bidi}}^2$ | $k_{\text{unidi}}^2$ | $k_{\text{bidi}}^1$ | $k_{\text{unidi}}^1$ |
|---|---|---|---|---|---|---|---|---|
| $k_{\text{bidi}}^4$ | 67.81±0.06 | 66.87±0.19 | 68.09±0.34 | 65.95±0.04 | 67.45±0.13 | 66.72±0.13 | 65.10±0.21 | 64.82±0.23 |
| $k_{\text{unidi}}^4$ | 61.02±0.23 | 77.49±0.21 | 72.43±0.02 | 78.41±0.06 | 74.75±0.38 | 78.19±0.23 | 77.00±0.36 | 77.87±0.34 |
| $k_{\text{bidi}}^3$ | 74.63±1.15 | 72.17±0.66 | 74.93±0.26 | 72.75±0.26 | 75.87±0.04 | 72.83±0.17 | 72.66±0.17 | 71.23±0.11 |
| $k_{\text{unidi}}^3$ | 57.73±0.49 | 77.82±0.30 | 72.55±0.19 | 77.96±0.13 | 76.48±0.11 | **79.22±0.15** | 77.38±0.66 | 77.74±0.04 |
| $k_{\text{bidi}}^2$ | 64.03±0.34 | 73.71±0.49 | 75.14±0.21 | 74.88±0.34 | 77.42±0.19 | 75.22±0.47 | 76.59±0.51 | 74.48±0.41 |
| $k_{\text{unidi}}^2$ | 52.58±0.04 | 74.97±0.68 | 68.80±0.53 | 77.51±0.02 | 73.30±0.56 | 78.90±0.17 | 76.08±0.38 | 76.33±0.09 |
| $k_{\text{bidi}}^1$ | 49.17±0.30 | 74.33±0.04 | 69.20±0.09 | 75.22±0.34 | 75.25±0.02 | 76.78±0.45 | 78.32±0.49 | 78.43±0.09 |
| $k_{\text{unidi}}^1$ | 34.05±0.01 | 72.08±0.79 | 55.75±0.21 | 74.63±0.26 | 66.19±0.02 | 76.91±0.28 | 75.50±0.53 | 78.15±0.02 |

Large-to-small: Transitioning from a broader to a narrower window results in improved model performance (e.g., from 4 to 3, or from 3 to 2). We attribute this phenomenon to the fact that training LLMs is more challenging from wider perspectives, where the model must contend with a greater amount of context and potential noise. When shifting to a narrower perspective, the model benefits from a reduced level of complexity and fewer distractions, leading to better.

**Overall(Q3)**, our empirical studies reveal that the Attention Window with connection information significantly impacts LLM's understanding of graph structure. Training with non-fully connected views containing certain topological link information aids LLMs in understanding graph structures and facilitates transfer from small to large perspectives.

**Other work in Appendix**

A detailed description of the setup is provided in **Ap-**

pendix A, with information about the dataset in Appendix B. Additional related work is discussed in Appendix C. Attention plots and experimental results for other datasets (base LLMs) are presented in **Appendices** H, G, I, and F.

## 4. Findings and Conclusion

**Findings.** We found that although LLMs gradually become aware of graph data during training, they fail to effectively leverage the connectivity information within these graphs. It was also discovered that fine-tuned LLMs possess the ability to capture relationships between nodes and text, but they lack the capability to model relationships among nodes. This limitation manifests as "Attention Sink" and "Skewed Line Sink" phenomena in attention interactions.

Additionally, we found that fully connected attention windows of LLMs are not suitable for training on graph data. A

better approach is to train under a smaller perspective that incorporates graph topology information and then perform transfer learning.

**Conclusion and Limitation.** We explored the attention mechanisms of LLMs on graph data and discovered numerous phenomena, providing directional guidance for further research by the graph learning community. At the same time, due to limitations in length and resources, we only analyzed a portion of the discovered phenomena. There remains a wealth of unexplored phenomena awaiting further investigation by the research community.

## Impact Statement

This paper presents work whose goal is to advance the field of machine learning. There are many potential societal consequences of our work, none of which we feel must be specifically highlighted here.

## Acknowledgements

This study was partially funded by the supports of National Natural Science Foundation of China (72471165) and Natural Science Foundation of Tianjin (No. 24JCQNJC01560).

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

# A. Implementation Details

Throughout the experiments, we maintained LLama2-7B (Touvron et al., 2023) as our base model and used an 8x8 scheme for neighbor sampling. For each dataset, during the experimental process, we processed the raw text from the datasets using the base model to serve as the embedding features for the nodes, treating each node akin to a token in the manner of LLama. Regarding the data collected as described in A.1, we gathered the attention scores for each node across different layers and heads, scaling them using a logarithmic function. For the perturbed data in A.2: (II) We performed random swaps between two nodes, repeating this procedure ten times consecutively. (IV) We conducted random exchanges of nodes within the same batch to introduce disorder. More details are shown in Table 5.

*Table 5.* Hyperparameters for the Roman-Empire, Amazon-Ratings, and Pubmed, Wikics.

| Hyperparameters | Amazon-Ratings | Pubmed | Wikcis | Roman |
|---|---|---|---|---|
| learning rate | 1e-4 | 1e-4 | 1e-4 | 1e-4 |
| warmup | 0.05 | 0.05 | 0.05 | 0.05 |
| gradient accumulation steps | 8 | 8 | 8 | 8 |
| batch size | 4 | 4 | 4 | 4 |
| epoch | 1 | 1 | 1 | 1 |
| num_beams | 2 | 2 | 2 | 2 |
| use_embedding | Ture | True | False | False |

# B. Datasets

When selecting datasets, we considered a broad spectrum and chose heterogeneous graph datasets Amazon-Rating (Platonov et al., 2023) and Roman-Empire (Platonov et al., 2023), as well as homogeneous graph datasets Pubmed and Wi-kiCS (Mernyei & Cangea, 2020). The statistical metrics of each dataset are shown in the following Table 6.

*Table 6.* Statistics of datasets

| | Roman-Empire | Amazon-Ratings | Wikics | Pubmed |
|---|---|---|---|---|
| nodes | 22,662 | 24,492 | 11,701 | 19,717 |
| edges | 32,927 | 93,050 | 216,123 | 44,338 |
| avg degree | 2.91 | 7.60 | 36.89 | 4.49 |
| node features | 4096 | 4096 | 4096 | 4096 |
| classes | 18 | 5 | 10 | 3 |
| edge homophily | 0.05 | 0.38 | - | - |
| adjusted homophily | -0.05 | 0.14 | - | - |

# C. Additional Related Work

Recent advancements have delved into leveraging Large Language Models (LLMs) within graph structure domains. Studies such as GLEM (Zhao et al., 2022), along with other works (Yang et al., 2021), have probed into the integration of LLMs and Graph Neural Networks (GNNs) through joint training frameworks. The approach taken by (He et al., 2023) employs LLMs to forecast node ranking classifications and offers comprehensive insights to enrich the quality of GNN embeddings. Meanwhile, Sun et al. (2023) has exploited LLMs for generating pseudo-labels aimed at enhancing the representation of graph topologies. Moreover, there has been an emphasis on advancing the direct processing capabilities of LLMs for textual graphs. For instance, InstructGLM (Ye et al., 2023) pioneers the use of instruction tuning based on LLMs to articulate graph structures and node characteristics, effectively addressing graph-related tasks. An interactive fusion of LLMs and GNNs is also presented by (Qiao et al., 2024). Despite these strides, LLMs face challenges when dealing with structured data that has been converted into natural language, frequently leading to less than optimal outcomes. To address this issue, LLaGA (Chen et al., 2024a) reformulates node-link information into sequential data, thereby applying instruction tuning that enhances LLM comprehension while preserving structural node information. UniGraph (He & Hooi, 2024) implements a masked strategy for co-training LLMs and GNNs together, achieving robust generalization across diverse graphs and

datasets. Additionally, Kong et al. (2024) investigates the creation of graph-based foundational models that merge LLMs with GNNs. GraphAdapter (Huang et al., 2024) utilizes a GNN model as an adapter working alongside LLMs for TAG tasks, which aids in task-specific fine-tuning via external access.

Moreover, recent efforts have increasingly focused on designing modules from a more comprehensive perspective to achieve better performance. In the context of Zhu et al. (2024a), mimicking the aggregation process of GNNs through instructions forces LLMs to learn how to aggregate node information effectively. The work by Zhang et al. (2024) employs LLMs to compress lengthy raw texts hierarchically, ultimately condensing them into a small number of tokens suitable for graph tasks. This hierarchical compression allows for efficient representation of complex textual data in a form that can be readily processed by graph-based models.

Zhang et al. (2024) approach incorporates the information from neighboring nodes when performing attention calculations on tokens within its own text, while also reducing the number of tokens. During the final inference phase, it refines predictions by re-evaluating the surrounding neighbor nodes, thereby enhancing the accuracy and relevance of the outcomes.

## D. Other Base Models for Disruption Performance

*Table 7.* Disruption Performance. Node classification results (accuracy(%) ±std) for 4 runs on four real-world datasets and five levels of disruption. The model used is **Vicuna-7B**, with node sampling configured as 8x8.

| Dataset | Raw | (I) | (II) | (III) | (IV) |
|---|---|---|---|---|---|
| Wikics | 0.7899± 0.004 | 0.7898± 0.002 | 0.7919± 0.001 | 0.7773± 0.002 | 0.7101± 0.005 |
| Pubmed | 0.8322±0.002 | 0.8345±0.001 | 0.8300±0.001 | 0.7434±0.003 | 0.7453±0.001 |
| Amazon-Ratings | 0.4541±0.001 | 0.4544±0.001 | 0.4498±0.002 | 0.4135±0.001 | 0.3943±0.001 |
| Roman-Empire | 0.8094±0.001 | 0.8001±0.001 | 0.7883±0.003 | 0.6543±0.001 | 0.6016±0.001 |

*Table 8.* Disruption Performance. Node classification results (accuracy(%) ±std) for 4 runs on four real-world datasets and five levels of disruption. The model used is **LLama3-7B**, with node sampling configured as 8x8.

| Dataset | Raw | (I) | (II) | (III) | (IV) |
|---|---|---|---|---|---|
| Wikics | 0.7988± 0.001 | 0.7981± 0.006 | 0.7988± 0.005 | 0.7487± 0.002 | 0.7209± 0.001 |
| Pubmed | 0.8448±0.001 | 0.8449±0.003 | 0.8442±0.003 | 0.8067±0.003 | 0.7753±0.002 |
| Amazon-Ratings | 0.4515±0.001 | 0.4523±0.002 | 0.3985±0.003 | 0.3889±0.002 | 0.3744±0.005 |
| Roman-Empire | 0.8108±0.002 | 0.8007±0.002 | 0.7922±0.002 | 0.6324±0.001 | 0.5618±0.002 |

## E. Statistical Analysis of Attention Score Distributions

It can be seen that most of them are consistent with our presentation and our conclusions.

*Table 9.* Statistical Analysis of Attention Score Distributions. *Note: ** indicates p-value ¡ 0.01, otherwise p ¿ 0.05 for t-test and KS test. JS Divergence values are provided as is.*

| Comparison | Amazon | | | WikiCS | | |
|---|---|---|---|---|---|---|
| | T-Test | KS Test | JS | T-Test | KS Test | JS |
| Before vs After (Text) | 95.066** | 0.118** | 0.0066 | 61.764** | 0.107** | 0.0038 |
| Before vs After (Node) | 69.334** | 0.116** | 0.0113 | 118.293** | 0.216** | 0.0307 |
| Node vs Text (Before) | −337.116** | 0.428** | 0.1423 | −268.998** | 0.362** | 0.0766 |
| Node vs Text (After) | −318.054** | 0.438** | 0.1123 | −184.605** | 0.285** | 0.0264 |

## F. Disruption Attention Score

In Figure 2, we illustrate the attention scores from neighboring nodes to the central node before and after training, without altering the structural information. Here, we show the attention scores from neighboring nodes to the central node under various perturbation conditions.

*Table 10.* Disruption Attention Score (Amazon-Ratings). The attention scores from neighboring nodes to the central node, were presented as mean values with standard deviations, using a 1:8 sampling ratio.

| Node | 1 | 2 | 3 | 4 | 5 | 6 | 7 | 8 |
|---|---|---|---|---|---|---|---|---|
| (raw) | 0.017±0.025 | 0.018±0.025 | 0.012±0.019 | 0.008±0.011 | 0.007±0.011 | 0.006±0.009 | 0.004±0.005 | 0.004±0.005 |
| (I) | 0.018±0.025 | 0.018±0.025 | 0.012±0.020 | 0.008±0.012 | 0.007±0.010 | 0.006±0.008 | 0.004±0.005 | 0.004±0.005 |
| (II) | 0.017±0.024 | 0.017±0.025 | 0.012±0.019 | 0.008±0.010 | 0.007±0.010 | 0.006±0.009 | 0.004±0.005 | 0.004±0.005 |
| (III) | 0.019±0.026 | 0.018±0.026 | 0.013±0.020 | 0.008±0.011 | 0.007±0.010 | 0.006±0.008 | 0.004±0.005 | 0.004±0.005 |
| (IV) | 0.017±0.025 | 0.018±0.025 | 0.012±0.019 | 0.008±0.011 | 0.007±0.011 | 0.006±0.009 | 0.004±0.005 | 0.004±0.005 |

*Table 11.* Disruption Attention Score (Wikics). The attention scores from neighboring nodes to the central node, were presented as mean values with standard deviations, using a 1:8 sampling ratio.

| Node | 1 | 2 | 3 | 4 | 5 | 6 | 7 | 8 |
|---|---|---|---|---|---|---|---|---|
| (raw) | 0.017±0.021 | 0.012±0.017 | 0.009±0.014 | 0.006±0.011 | 0.006±0.011 | 0.005±0.010 | 0.004±0.009 | 0.004±0.009 |
| (I) | 0.018±0.022 | 0.013±0.018 | 0.009±0.015 | 0.007±0.013 | 0.006±0.012 | 0.005±0.012 | 0.004±0.011 | 0.004±0.010 |
| (II) | 0.017±0.020 | 0.012±0.016 | 0.008±0.013 | 0.006±0.010 | 0.006±0.010 | 0.005±0.009 | 0.004±0.008 | 0.004±0.007 |
| (III) | 0.016±0.019 | 0.011±0.013 | 0.007±0.009 | 0.005±0.006 | 0.005±0.006 | 0.004±0.005 | 0.003±0.004 | 0.003±0.003 |
| (IV) | 0.018±0.022 | 0.013±0.018 | 0.009±0.015 | 0.007±0.012 | 0.006±0.012 | 0.005±0.011 | 0.004±0.010 | 0.004±0.010 |

*Table 12.* Disruption Attention Score (Roman). The attention scores from neighboring nodes to the central node, were presented as mean values with standard deviations, using a 1:8 sampling ratio.

| Node | 1 | 2 | 3 | 4 | 5 | 6 | 7 | 8 |
|---|---|---|---|---|---|---|---|---|
| (raw) | 0.023±0.021 | 0.016±0.016 | 0.014±0.014 | 0.012±0.012 | 0.010±0.010 | 0.009±0.010 | 0.011±0.011 | 0.004±0.002 |
| **(I)** | 0.024±0.021 | 0.016±0.016 | 0.013±0.013 | 0.011±0.011 | 0.009±0.010 | 0.009±0.009 | 0.011±0.011 | 0.004±0.003 |
| **(II)** | 0.024±0.021 | 0.016±0.016 | 0.014±0.014 | 0.012±0.013 | 0.010±0.011 | 0.010±0.010 | 0.011±0.011 | 0.004±0.003 |
| **(III)** | 0.022±0.020 | 0.015±0.014 | 0.011±0.011 | 0.010±0.011 | 0.009±0.009 | 0.008±0.008 | 0.006±0.010 | 0.004±0.001 |
| **(IV)** | 0.023±0.021 | 0.016±0.016 | 0.014±0.014 | 0.012±0.012 | 0.010±0.010 | 0.009±0.010 | 0.011±0.012 | 0.004±0.003 |

# G. Attention Score Matrix(Tokens)

## G.1. Token to Token

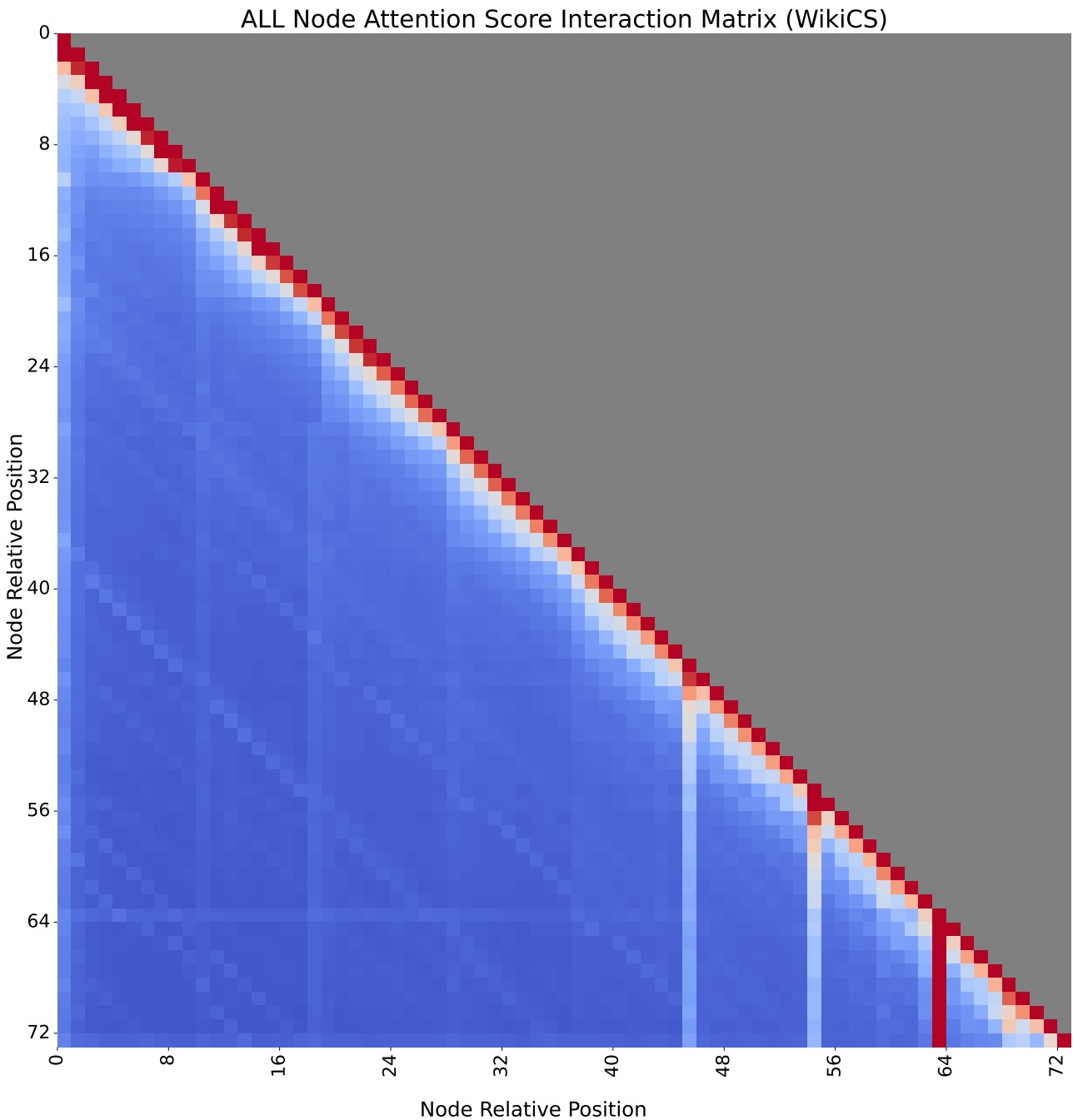

*Figure 7.* Attention score interaction matrix(Nodes) in Wikics.

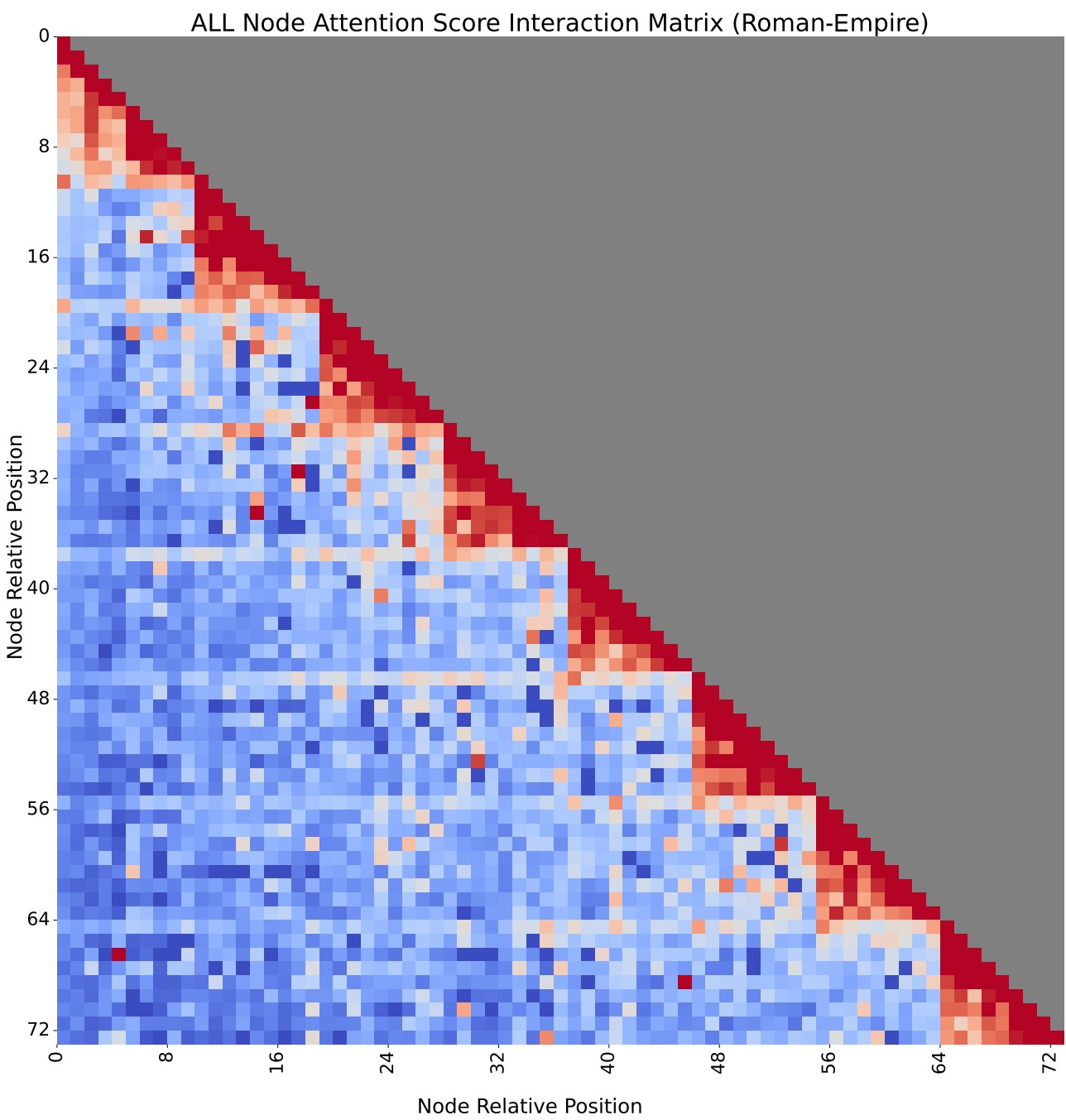

*Figure 8.* Attention score interaction matrix(Nodes) in Roman-Empire.

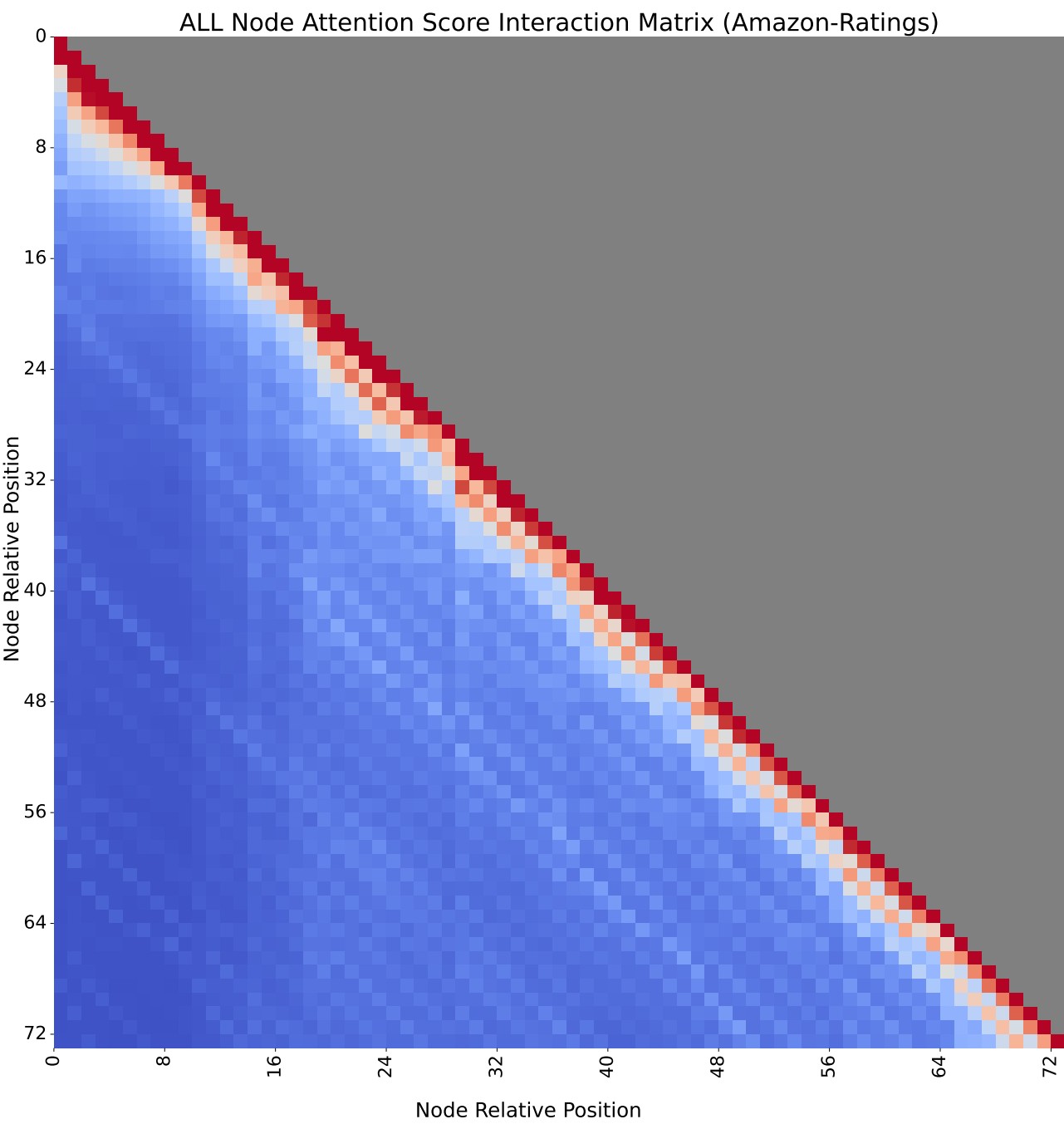

*Figure 9.* Attention score interaction matrix(Nodes) in Amazon-Ratings.

**G.2. Text to Text**

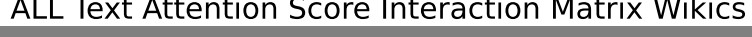

Figure 10. Attention score interaction matrix(Text) in Wikics.

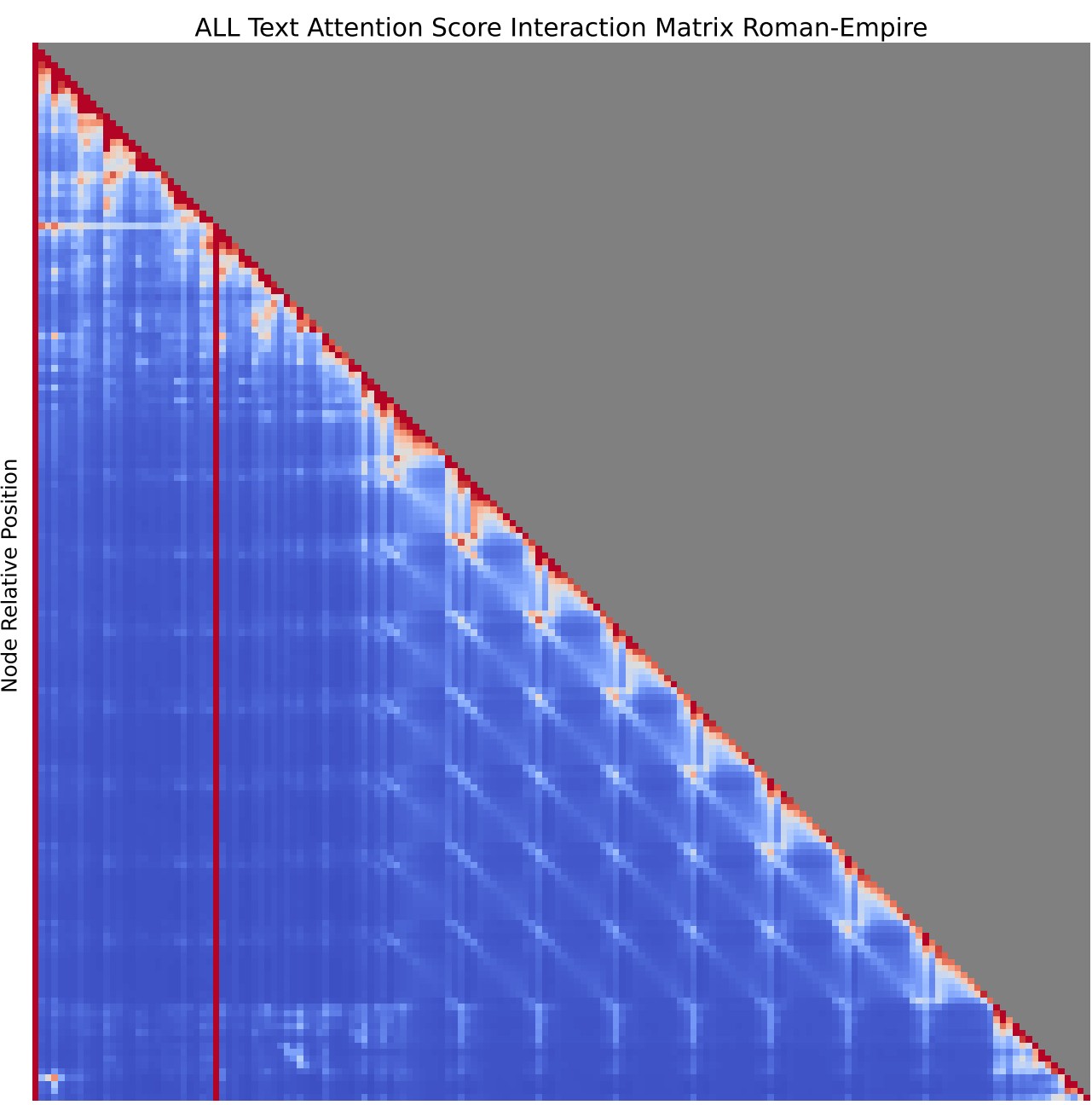

*Figure 11.* Attention score interaction matrix(Text) in Roman-Empire.

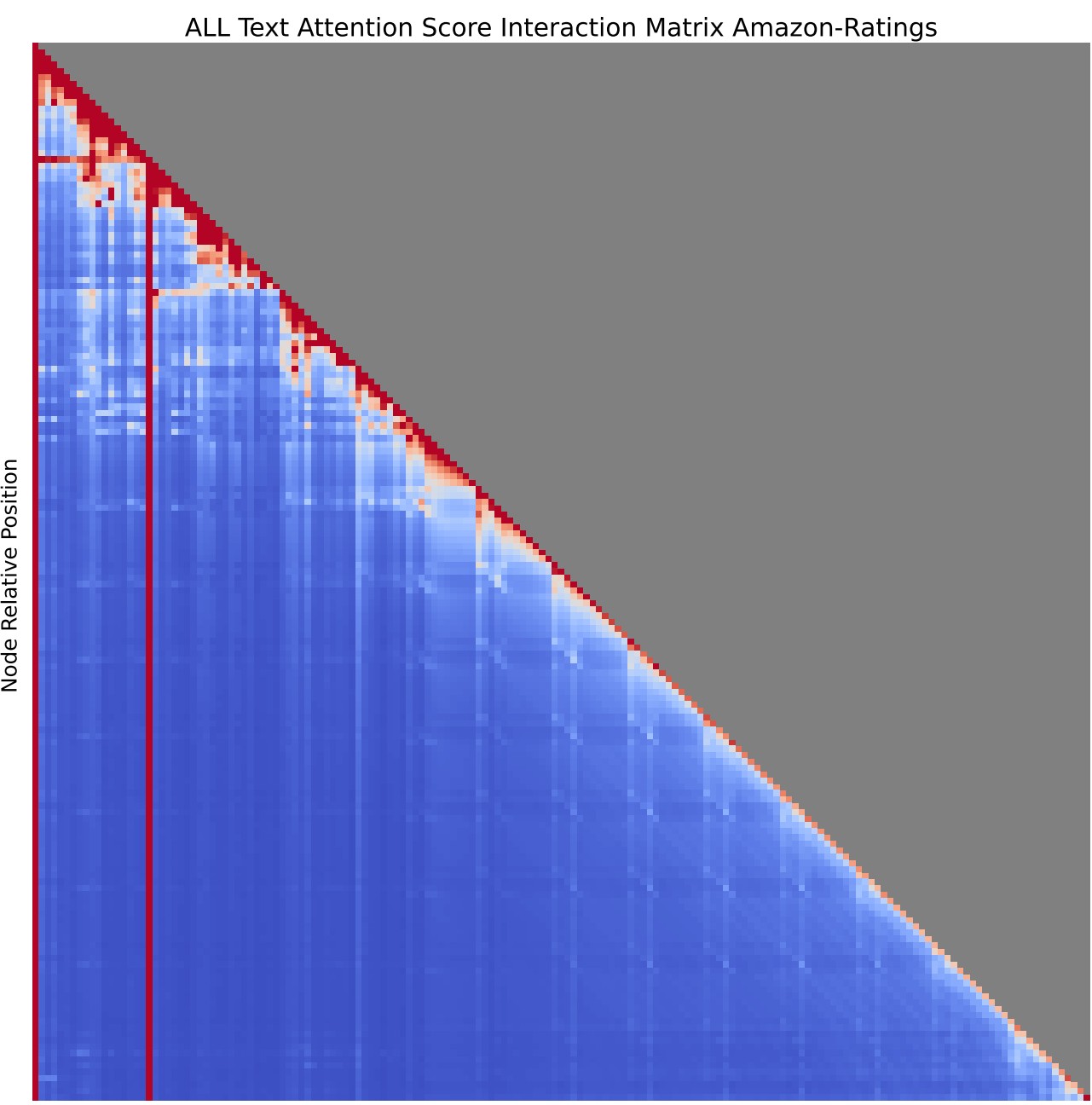

*Figure 12.* Attention score interaction matrix(Text) in Amazon-Ratings.

## H. Attention Score among First Nodes(with child nodes)

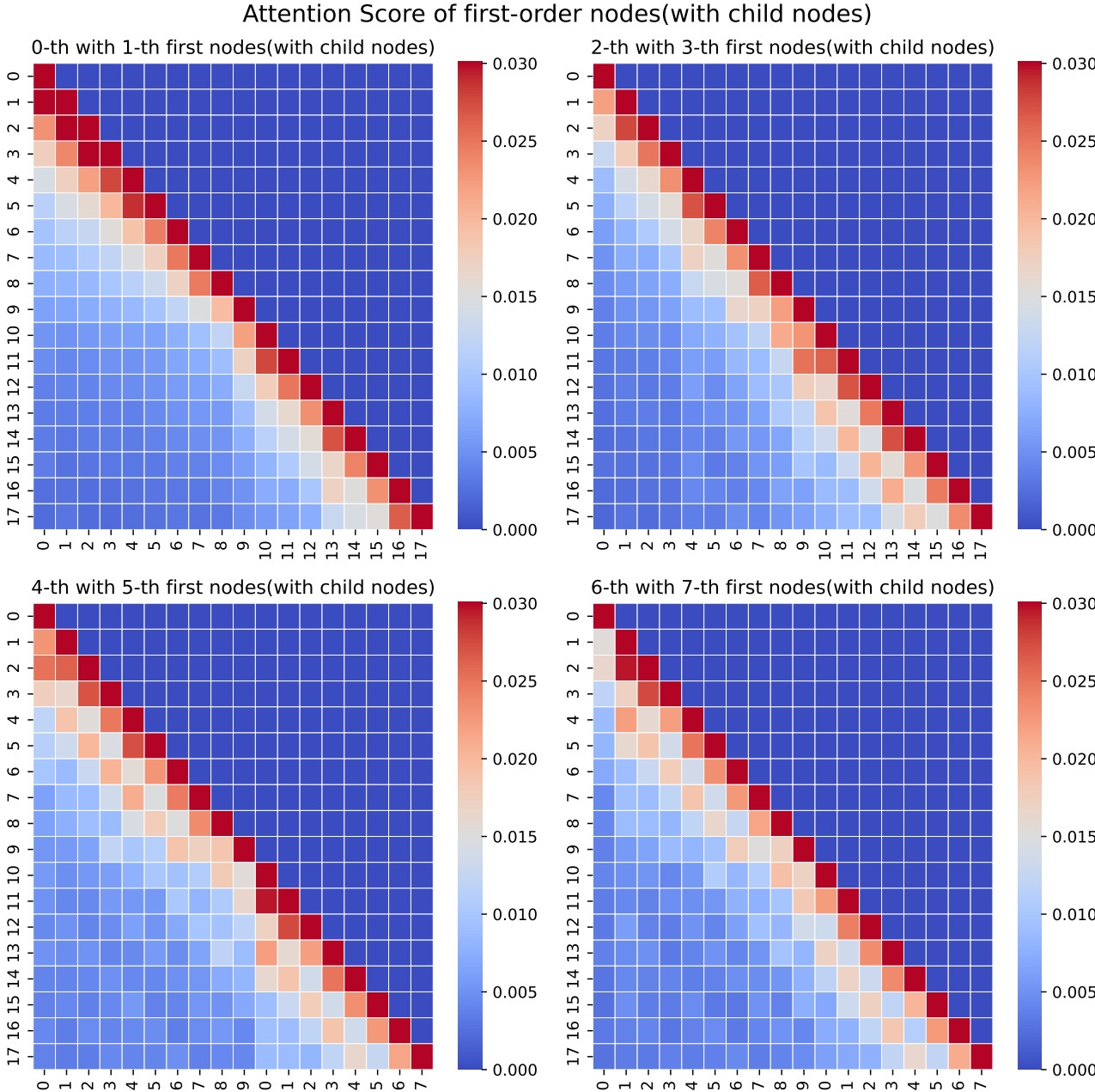

*Figure 13.* Visualization of the average attention(Attention Score among First Nodes(with child nodes)) in Amazon-Ratings((1+8)*2).

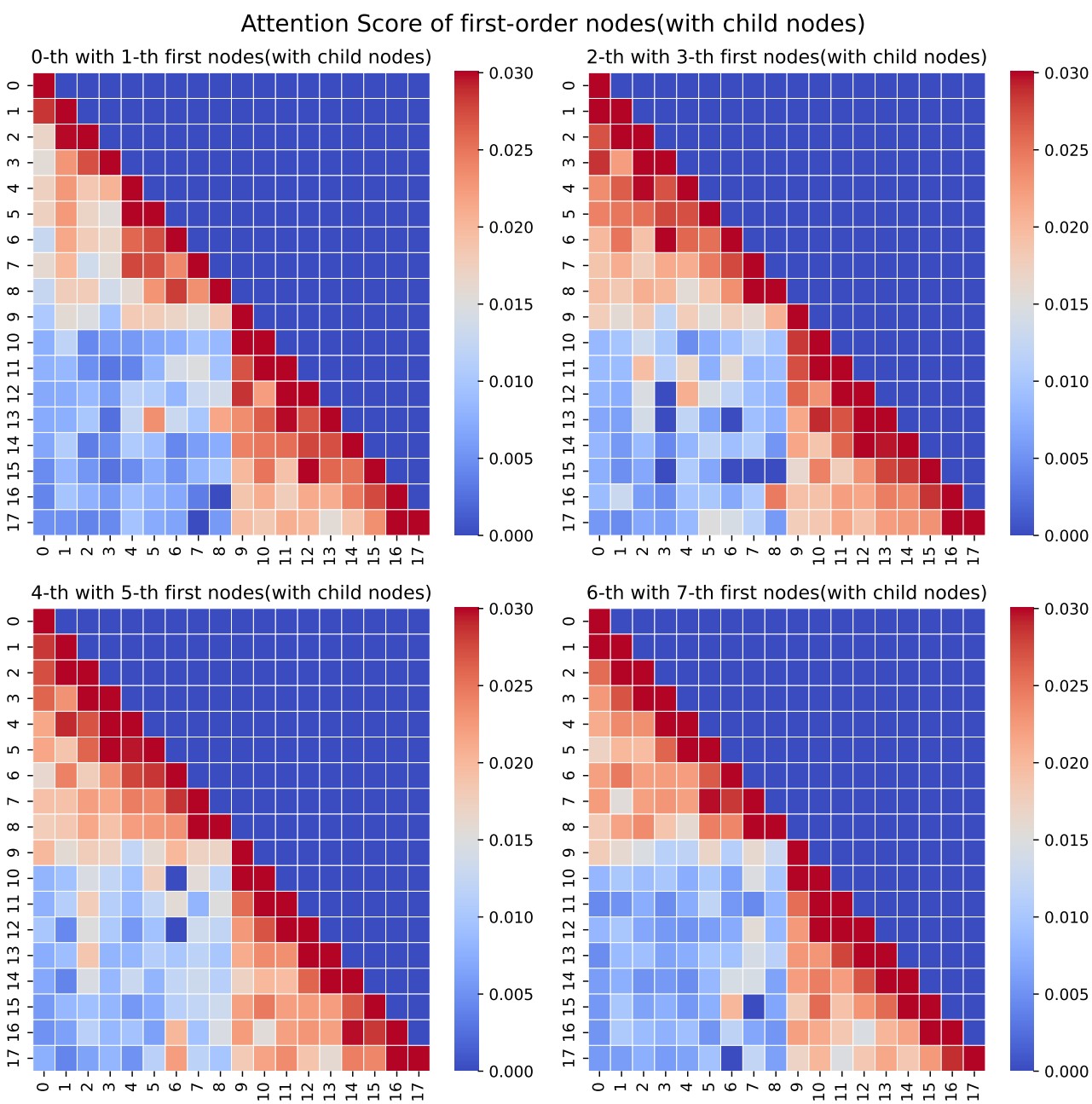

*Figure 14.* Visualization of the average attention(Attention Score among First Nodes(with child nodes)) in Roman-Empire((1+8)*2).

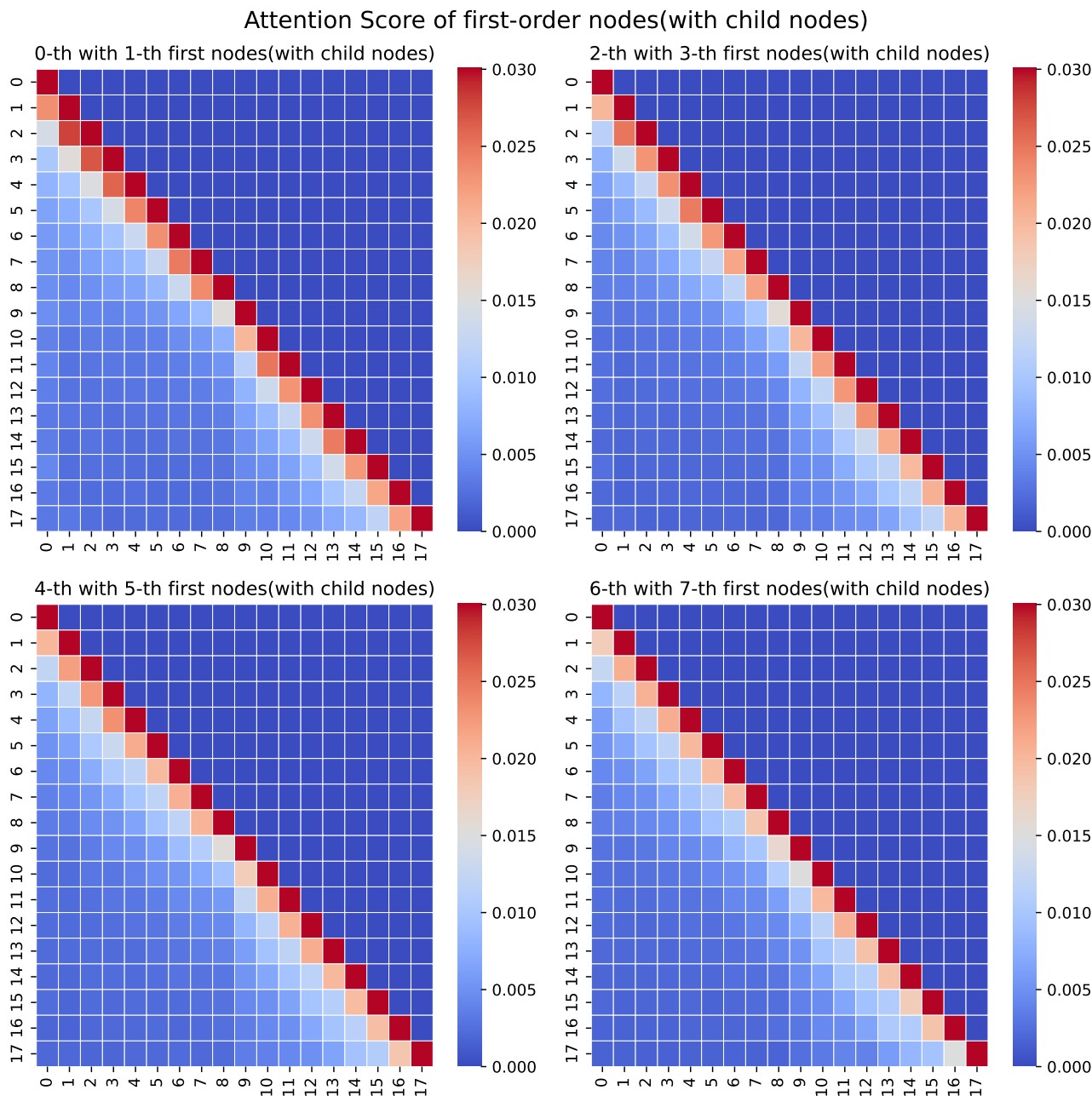

*Figure 15.* Visualization of the average attention(Attention Score among First Nodes(with child nodes)) in Wikics((1+8)*2).

## I. Different Layer Attention Score between First Nodes and child nodes

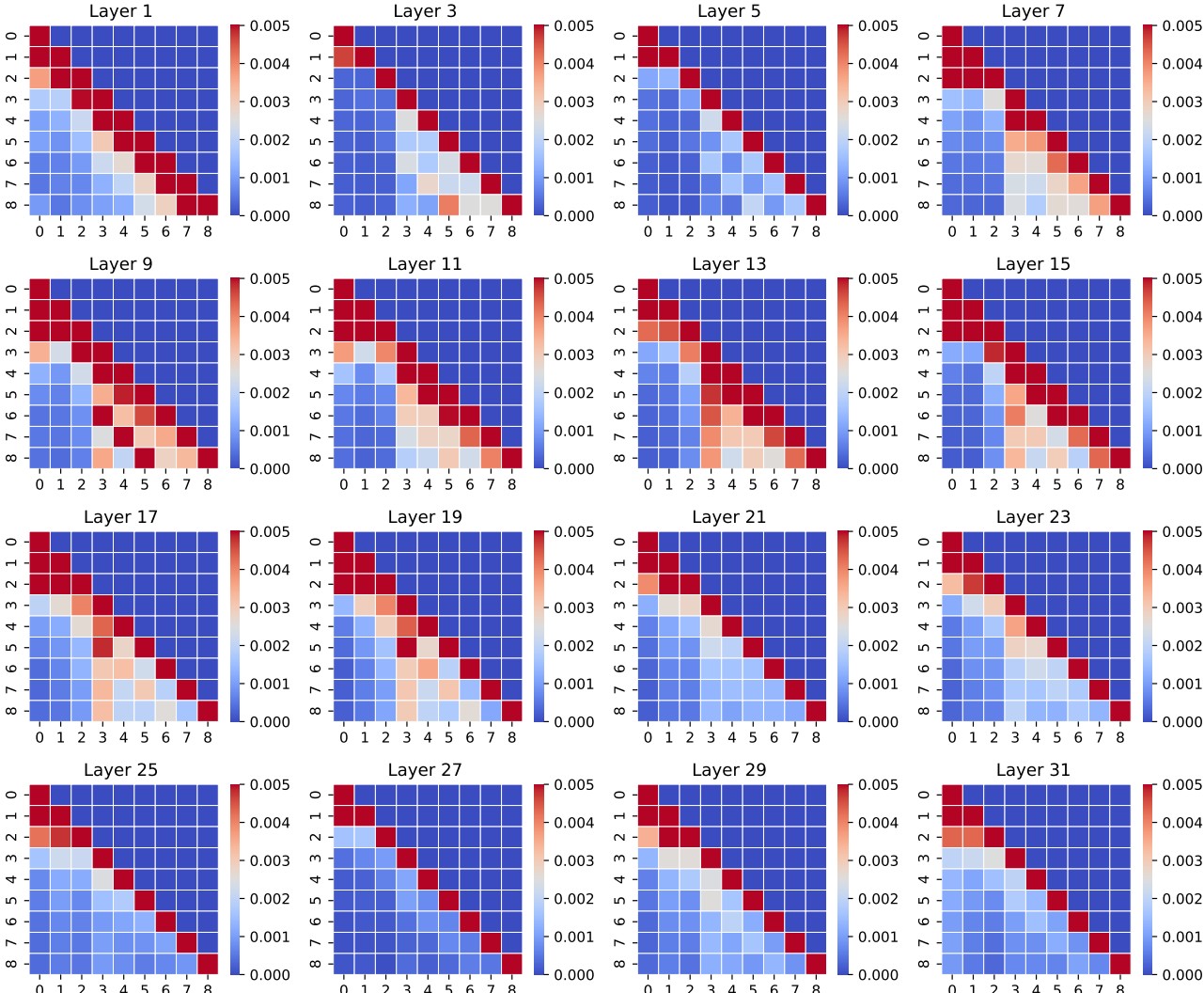

*Figure 16.* Visualization of the average attention(Center nodes and First-order nodes) in Amazon-Ratings.

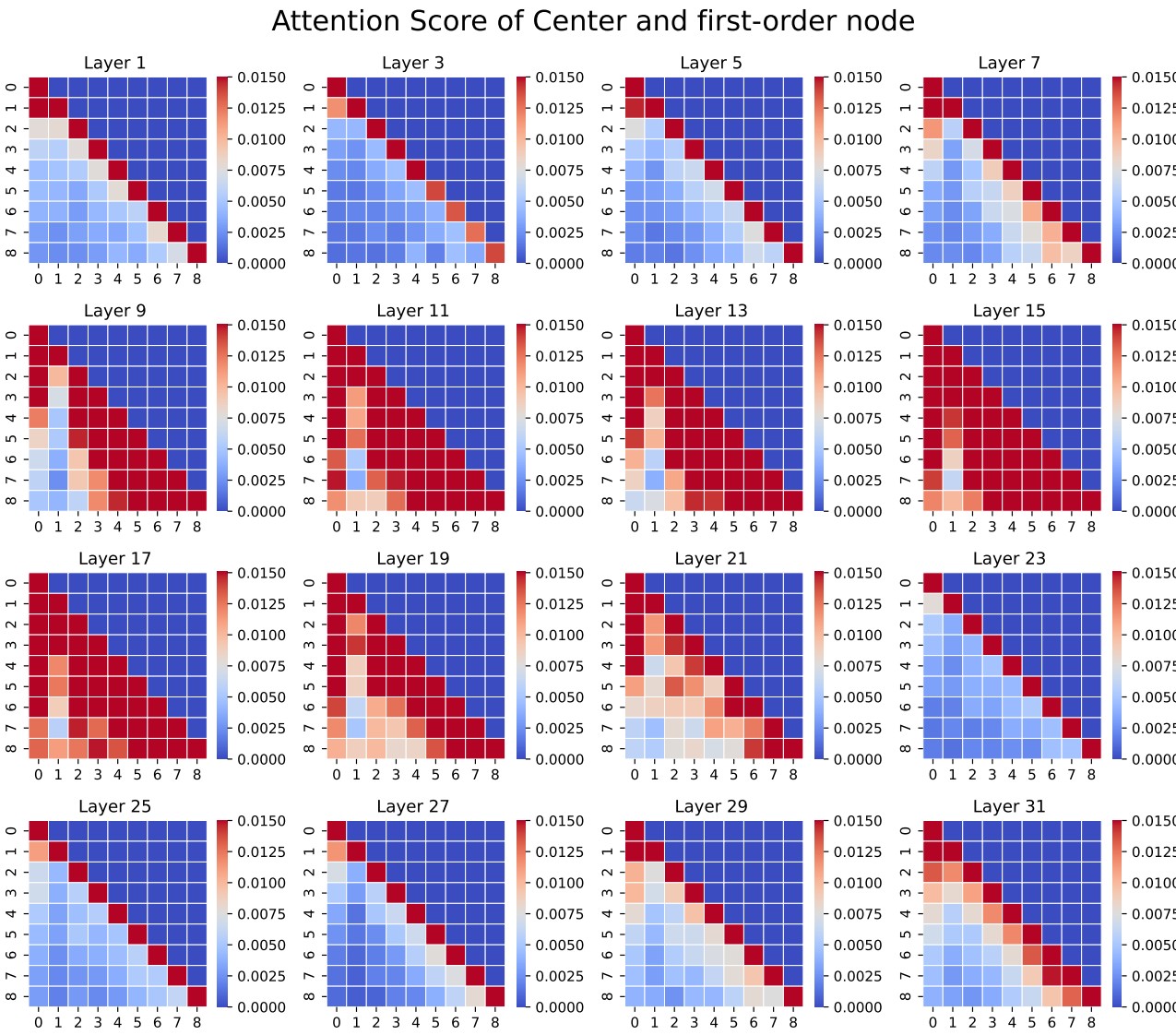

*Figure 17.* Visualization of the average attention(Center nodes and First-order nodes) in Roman-Empire.

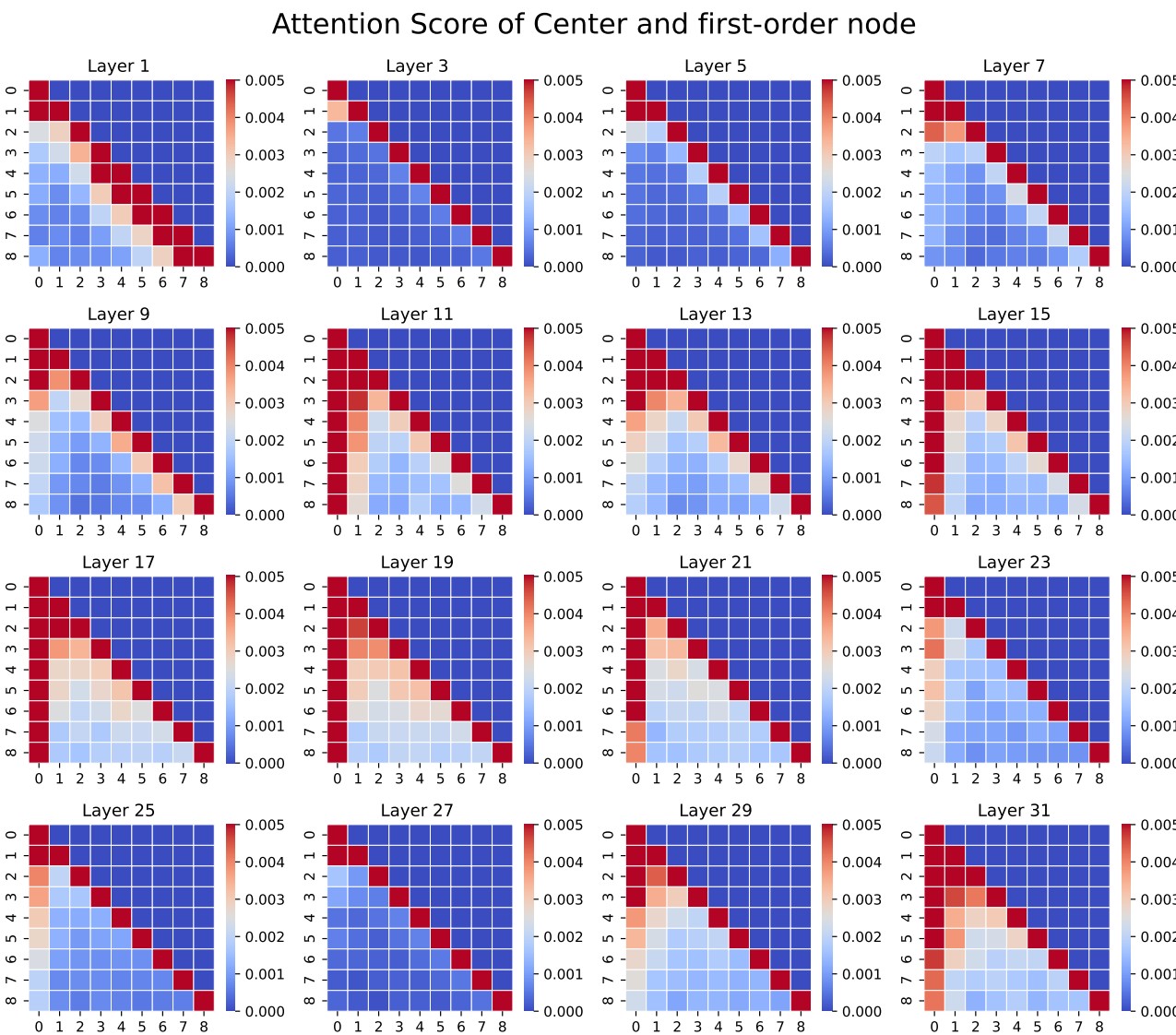

*Figure 18.* Visualization of the average attention(Center nodes and First-order nodes) in Wikics.

## J. Prompt

Here, we list all the prompts we used in this paper on different datasets, we use the following prompt:.

- **Roman-Empire:** "<User >: In an article, words that have dependency relationships (where one word depends on another) are connected, forming a dependency graph. Based on the connections between words, determine the syntactic role of each word. Given that a word [] that connect [], what is the word [] syntactic role? <Assistant >: "

- **Amazon-Ratings:** "<User >: In a product graph dataset, edges connect products that are frequently purchased together. Based on the connections between products (books, music CDs, DVDs, VHS tapes), predict the average rating given by reviewers for the products. Given that a product [] that connect [], what is the product [] rating? <Assistant >: "

- **Pubmed:** "<User >: In medical paper dataset, papers that cite each other form a linkage relationship. Based on the linkage relationships among papers, the research directions of medical papers can be predicted. Given that a paper [] that connect [], What is the category of the paper []? <Assistant >:: "

- **Wikics:** "<User >: In paper dataset, papers that cite each other form a linkage relationship. Based on the linkage relationships among papers, the research directions of papers can be predicted. Given that a paper [] that connect [], What is the category of the paper []? <Assistant >:: "

## K. Different Templates

We designed various Templates for analysis and discussion, inspired by the the paper (TALK LIKE A GRAPH).

The results of the disruption experiment are as follows. From the Table K, it can be seen that although different templates have varying model performances, the overall trend is consistent, showing utilization of link information specifically on the Roman dataset, while link information did not play a role on most other datasets. Our initial template (1) remains the best performing and makes the best use of link information.

*Table 13.* Comparison of different datasets under various templates.

| Dataset | Raw | (I) | (II) | (III) | (IV) |
|---|---|---|---|---|---|
| **(1)** | | | | | |
| Wikics | $0.7862 \pm 0.007$ | $0.7847 \pm 0.004$ | $0.7907 \pm 0.0035$ | $0.7670 \pm 0.003$ | $0.7087 \pm 0.005$ |
| Pubmed | $0.8367 \pm 0.003$ | $0.8363 \pm 0.001$ | $0.8364 \pm 0.002$ | $0.7698 \pm 0.003$ | $0.7835 \pm 0.001$ |
| Amazon-Ratings | $0.4486 \pm 0.002$ | $0.3980 \pm 0.003$ | $0.3977 \pm 0.002$ | $0.3915 \pm 0.003$ | $0.3813 \pm 0.001$ |
| Roman-Empire | $0.8089 \pm 0.001$ | $0.7918 \pm 0.001$ | $0.7910 \pm 0.002$ | $0.6290 \pm 0.002$ | $0.5784 \pm 0.002$ |
| **(2)** | | | | | |
| Wikics | $0.7795 \pm 0.006$ | $0.7789 \pm 0.005$ | $0.7778 \pm 0.003$ | $0.7512 \pm 0.004$ | $0.6871 \pm 0.006$ |
| Pubmed | $0.8112 \pm 0.004$ | $0.8108 \pm 0.002$ | $0.8050 \pm 0.001$ | $0.7745 \pm 0.002$ | $0.7783 \pm 0.002$ |
| Amazon-Ratings | $0.4231 \pm 0.003$ | $0.4038 \pm 0.002$ | $0.3923 \pm 0.003$ | $0.3869 \pm 0.002$ | $0.3768 \pm 0.002$ |
| Roman-Empire | $0.7932 \pm 0.002$ | $0.7961 \pm 0.002$ | $0.7935 \pm 0.001$ | $0.7228 \pm 0.003$ | $0.6130 \pm 0.001$ |
| **(3)** | | | | | |
| Wikics | $0.7806 \pm 0.007$ | $0.7793 \pm 0.004$ | $0.7851 \pm 0.0035$ | $0.7618 \pm 0.003$ | $0.7025 \pm 0.005$ |
| Pubmed | $0.8315 \pm 0.003$ | $0.8310 \pm 0.001$ | $0.8312 \pm 0.002$ | $0.7643 \pm 0.003$ | $0.7781 \pm 0.001$ |
| Amazon-Ratings | $0.4438 \pm 0.002$ | $0.3932 \pm 0.003$ | $0.3929 \pm 0.002$ | $0.3867 \pm 0.003$ | $0.3765 \pm 0.001$ |
| Roman-Empire | $0.8030 \pm 0.001$ | $0.7864 \pm 0.001$ | $0.7856 \pm 0.002$ | $0.6234 \pm 0.002$ | $0.5728 \pm 0.002$ |
| **(4)** | | | | | |
| Wikics | $0.7811 \pm 0.006$ | $0.7833 \pm 0.005$ | $0.7812 \pm 0.0030$ | $0.7665 \pm 0.004$ | $0.7092 \pm 0.006$ |
| Pubmed | $0.8325 \pm 0.004$ | $0.8322 \pm 0.002$ | $0.8317 \pm 0.001$ | $0.7897 \pm 0.002$ | $0.7064 \pm 0.002$ |
| Amazon-Ratings | $0.4379 \pm 0.003$ | $0.4085 \pm 0.002$ | $0.4072 \pm 0.003$ | $0.3978 \pm 0.002$ | $0.3885 \pm 0.002$ |
| Roman-Empire | $0.8091 \pm 0.002$ | $0.7915 \pm 0.002$ | $0.7913 \pm 0.001$ | $0.6285 \pm 0.003$ | $0.5788 \pm 0.001$ |

# L. Transfer Performance of Models across Different $k$ Values.

*Table 14.* Transfer Performance of Models across Different $k$ Values(Roman). **Rows** indicate the $k$ value used during training, **Columns** represent the $k$ value utilized during testing. The best performance for each $k$ value transfer is highlighted in **gray**, with the overall best performance **bolded**.

| | k=4 | | k=3 | | k=2 | | k=1 | |
| --- | --- | --- | --- | --- | --- | --- | --- | --- |
| | **bidi** | **unidi** | **bidi** | **unidi** | **bidi** | **unidi** | **bidi** | **unidi** |
| $k_{\text{bidi}}^{4}$ | 81.38±0.00 | 73.72±0.04 | 80.96±0.02 | 73.84±0.04 | 79.82±0.01 | 74.28±0.06 | 79.10±0.00 | 73.91±0.04 |
| $k_{\text{unidi}}^{4}$ | 68.51±0.04 | 80.73±0.00 | 70.51±0.04 | 80.37±0.01 | 73.04±0.04 | 79.66±0.01 | 72.48±0.01 | 79.73±0.01 |
| $k_{\text{bidi}}^{3}$ | 80.96±0.02 | 75.96±0.05 | 81.61±0.00 | 75.67±0.01 | 80.80±0.02 | 75.60±0.01 | 78.64±0.02 | 74.93±0.08 |
| $k_{\text{unidi}}^{3}$ | 67.41±0.06 | 81.79±0.01 | 69.50±0.05 | 81.68±0.02 | 72.62±0.01 | 81.60±0.01 | 72.74±0.03 | 81.59±0.02 |
| $k_{\text{bidi}}^{2}$ | 78.65±0.01 | 76.09±0.03 | 79.70±0.00 | 76.10±0.00 | 82.05±0.02 | 75.76±0.01 | 81.26±0.00 | 75.94±0.01 |
| $k_{\text{unidi}}^{2}$ | 67.75±0.03 | 82.66±0.01 | 69.71±0.04 | 82.63±0.02 | 74.37±0.02 | 82.87±0.01 | 76.54±0.03 | 82.11±0.02 |
| $k_{\text{bidi}}^{1}$ | 77.30±0.04 | 77.26±0.01 | 78.25±0.03 | 77.44±0.02 | 79.92±0.02 | 77.37±0.00 | 83.05±0.01 | 77.89±0.04 |
| $k_{\text{unidi}}^{1}$ | 64.81±0.05 | 82.54±0.01 | 66.55±0.01 | 82.56±0.00 | 71.17±0.01 | 82.78±0.01 | 73.53±0.00 | 83.12±0.01 |

*Table 15.* Transfer Performance of Models across Different $k$ Values(Amazon-Ratings). **Rows** indicate the $k$ value used during training, **Columns** represent the $k$ value utilized during testing. The best performance for each $k$ value transfer is highlighted in **gray**, with the overall best performance **bolded**.

| | k=4 | | k=3 | | k=2 | | k=1 | |
| --- | --- | --- | --- | --- | --- | --- | --- | --- |
| | **bidi** | **unidi** | **bidi** | **unidi** | **bidi** | **unidi** | **bidi** | **unidi** |
| $k_{\text{bidi}}^{4}$ | 43.79±0.43 | 35.99±0.51 | 42.26±0.11 | 35.19±0.11 | 38.81±0.16 | 32.81±0.21 | 30.08±0.07 | 28.98±0.07 |
| $k_{\text{unidi}}^{4}$ | 41.27±0.16 | 45.19±0.20 | 44.36±0.08 | 44.95±0.08 | 45.43±0.19 | 44.59±0.18 | 44.47±0.07 | 42.97±0.36 |
| $k_{\text{bidi}}^{3}$ | 45.28±0.01 | 39.31±0.33 | 45.63±0.08 | 38.46±0.47 | 43.56±0.02 | 35.24±0.52 | 35.55±0.25 | 29.10±0.20 |
| $k_{\text{unidi}}^{3}$ | 43.20±0.15 | 45.66±0.11 | 44.64±0.09 | 45.46±0.04 | 45.30±0.01 | 45.11±0.16 | 44.78±0.02 | 42.83±0.09 |
| $k_{\text{bidi}}^{2}$ | 44.31±0.10 | 44.40±0.04 | 45.09±0.05 | 44.24±0.03 | 45.18±0.20 | 43.66±0.04 | 42.78±0.11 | 41.42±0.16 |
| $k_{\text{unidi}}^{2}$ | 40.45±0.16 | 44.33±0.12 | 41.24±0.03 | 44.40±0.20 | 42.84±0.28 | 44.67±0.05 | 43.46±0.10 | 43.01±0.06 |
| $k_{\text{bidi}}^{1}$ | 32.02±0.09 | 42.30±0.11 | 37.66±0.11 | 44.63±0.07 | 40.68±0.03 | 45.14±0.00 | 46.17±0.03 | 45.30±0.34 |
| $k_{\text{unidi}}^{1}$ | 41.73±0.15 | 42.96±0.09 | 42.63±0.10 | 44.33±0.17 | 43.08±0.19 | 44.76±0.02 | 46.27±0.08 | 45.95±0.06 |

