# OpenReview forum: "Attention Mechanisms Perspective: Exploring LLM Processing of Graph-Structured Data"
_ICML.cc/2025/Conference — ICML 2025 poster_

### Official Review · Reviewer_yYED · 2025-03-09

**Overall Recommendation:** 4

**Summary:**

As an experimental analysis work, the article examines the limitations of LLMs in processing graph-structured data from the perspective of attention mechanisms. The experimental design is relatively comprehensive, covering changes in attention distribution, structural information interference, comparisons of different attention windows, etc., providing new insights into the application of LLMs in graph learning.

## update after rebuttal

The authors have addressd my all concerns and I have reviewed the comments from the other three reviewers. So, I keep my positive score.

**Claims And Evidence:**

The claims made in the submission can be supported by clear and convincing experiments, which provide different levels of validation on the proposed method.

**Essential References Not Discussed:**

Essential References are discussed in the paper.

**Experimental Designs Or Analyses:**

I have checked the soundness/validity of experimental designs or analyses in the submission. Many experiments on benchmark datasets show the effectiveness and superiority of the proposed method.

**Methods And Evaluation Criteria:**

The proposed methods and/or evaluation criteria make sense for the analysis of LLM attention on graph-structured data.

**Other Comments Or Suggestions:**

I have not any other comments or suggestions here. I have given all my comments in the “Other Strengths And Weaknesses”.

**Other Strengths And Weaknesses:**

Strength:
(1) The starting point of the problem is quite novel, focusing on the attention interaction of LLMs under graph structures, which is a particularly interesting aspect.
(2)Compared with existing works on LLM for Graph, most of which propose new architectures or methods, this article reveals and discovers interesting phenomena of the attention mechanism in graph-structured data through experimental analysis, offering better guiding value for future work.
(3) Detailed visualizations, including attention score interaction matrices and other charts, provide strong visual support.
(4)Discovery of important phenomena and issues: The article identifies "Attention Sink" and unique "Skewed Line Sink" phenomena of the LLM's attention mechanism under graph structures, potentially uncovering new research directions for LLM for Graph.

Weakness:
(1)Descriptions of the specific methods for calculating attention scores and conducting perturbation operations are not detailed enough. Providing more details or pseudocode would help readers understand the processes better.
(2)Lack of relevant citations or materials to prove certain points; for instance, the bimodal trend mentioned in the text focuses on mean differences with t-tests while KS tests emphasize differences between overall distributions ("Given that the KS test focuses on the overall distribution and the t-test emphasizes the mean value:"). It would be better if relevant materials were provided as evidence.

**Questions For Authors:**

(1)In the Attention Window section of the paper, the authors mention training with k not equal to 4 and then transferring to an attention window where k=4. When training with k≠4, will it introduce additional computational time consumption?
(2)The author designed a link information perturbation experiment (which is very interesting)， where the setup was based on gradually disrupting normal links. I wonder if there could be or if it's possible to add settings that enhance the link information?

**Relation To Broader Scientific Literature:**

The key contributions are: the authors analysis of LLM attention on graph-structured data. To my knowledge, this is the first empirical study to investigate LLMs for graph machine learning from the perspective of attention.

**Theoretical Claims:**

There are no theoretical claims included in the paper.

---

> ### Author Rebuttal · Authors · 2025-03-31
>
> >the bimodal trend mentioned in the text focuses on mean differences with t-tests while KS tests emphasize differences between overall distributions ("Given that the KS test focuses on the overall distribution and the t-test emphasizes the mean value:"). It would be better if relevant materials were provided as evidence.
>
> Thank you for your suggestion. In paper [1], the KS test is compared with the Anderson-Darling test (both are distribution-based tests), pointing out that the KS test is sensitive to global differences in distributions, whereas the t-test focuses only on mean-based hypotheses.
>
> [1] Chakraborti, S., & Lim, J. (2010). Comparing Distributions: The Two-Sample Anderson-Darling Test vs. the Kolmogorov-Smirnov Test. Journal of Applied Statistics, 37(1), 53–65.
>
> >In the Attention Window section of the paper, the authors mention training with k≠4 and then transferring to an attention window where k=4. When training with k≠4, will it introduce additional computational time consumption?
>
> Thank you for your consideration. In our model's operation, we only modified the computation of the **attention_mask** by adding some masking (similar to causal masking). This has a negligible impact on the model's training speed.
>
>
>
>
>
> >The author designed a link information perturbation experiment (which is very interesting)， where the setup was based on gradually disrupting normal links. I wonder if there could be or if it's possible to add settings that enhance the link information?
>
> First, your suggestion and idea are both highly feasible and significant. We have added an enhancement mode (V) to the perturbation experiment. Mode (V) uses a pre-trained GNN model to perform initial node filtering.
>
> The specific results are as follows:
>
>
> | Dataset       | Raw        | (I)        | (II)       | (III)      | (IV)       | (V)        |
> |---------------|------------|------------|------------|------------|------------|------------|
> | Wikics        | 0.7862 ± 0.007 | 0.7847 ± 0.004 | 0.7907 ± 0.0035| 0.7670 ± 0.003 | 0.7087 ± 0.005| 0.7897 ± 0.004     |
> | Pubmed        | 0.8367 ± 0.003 | 0.8363 ± 0.001 | 0.8364 ± 0.002 | 0.7698 ± 0.003 | 0.7835 ± 0.001| 0.8347 ± 0.002     |
> | Amazon-Ratings| 0.4486 ± 0.002 | 0.3980 ± 0.003 | 0.3977 ± 0.002 | 0.3915 ± 0.003 | 0.3813 ± 0.001| 0.4744 ± 0.004 |
> | Roman-Empire  | 0.8089 ± 0.001 | 0.7918 ± 0.001 | 0.7910 ± 0.002 | 0.6290 ± 0.002 | 0.5784 ± 0.002| 0.8177 ± 0.001     |
>
>
> The experimental setup for (V) may require further consideration, and we will conduct more repeated experiments and validations in the future. Thank you for your valuable suggestions regarding the perturbation experiment.
>
>
> >Descriptions of the specific methods for calculating attention scores and conducting perturbation operations are not detailed enough.
>
> Thank you for considering the reproducibility of the experiments. We will provide preliminary explanations here.
>
> Calculating attention scores: In our padding operation, we chose to discard the calculation of attention scores for pad tokens because this does not contribute to node attention calculations. Additionally, to avoid potential interference, we directly masked these tokens in the attention_mask.
>
> conducting perturbation operations：(II) Exchanging random numbers of child nodes between first-order nodes and further disrupting the information structure: Unlike (I), this operation involves transferring a random subset of child nodes between first-order nodes. Additionally, it introduces further disruptions to the overall structure.
>
> (III) Randomly shuffling the positions of first-order and second-order nodes, even allowing original second-order nodes to become first-order nodes: This operation randomizes the hierarchy and positions of nodes, potentially altering their roles (e.g., promoting second-order nodes to first-order status).

---

### Official Review · Reviewer_pYJ8 · 2025-03-13

**Overall Recommendation:** 5

**Summary:**

This article explores the attention mechanisms of Large Language Models (LLMs) when processing graph-structured data. It first observes that while LLMs are aware of graph-structured data, they fail to properly utilize link information. The analysis and explanation focus on two broad aspects: attention distribution and attention window. Several instructive conclusions or phenomena were found or concluded, such as LLMs' ability to capture relationships between nodes and text but lacking the capability to model relationships between nodes, like the unique Attention sink phenomenon in graph structures.

## update after rebuttal
The authors have addressed my concerns.

**Claims And Evidence:**

Yes.

**Essential References Not Discussed:**

None

**Experimental Designs Or Analyses:**

Yes, I have reviewed multiple experiments and figures. For instance, the method presented in Table 4, which involves training with a narrow field of view and testing with a wide field of view.

**Methods And Evaluation Criteria:**

YES, A variety of datasets, including heterogeneous, heterophilic, and text graphs, were selected for the experimental analysis, employing multiple evaluation criteria.

**Other Comments Or Suggestions:**

It is hoped that the author will make revisions and improvements based on the weaknesses and questions raised.

**Other Strengths And Weaknesses:**

Strengths:
Innovative experimental design; the study assesses the ability of LLMs to correctly utilize graph structure through methods that disrupt connection information at different levels, which is novel and clever, confirming the inadequacy of LLMs in utilizing topological information.

Discover multiple new phenomena and conclusions , contributing to the graph machine learning community.

The article explores the effectiveness of LLMs in handling graph data under different global visibility perspectives, and proposes a method of training under an intermediate state attention window before transferring to testing under a normal causal mask window. This approach balances the efficiency of the model and the feasibility of practical deployment.

Various statistical hypothesis methods (t-test, KS test, JS divergence) are used to evaluate changes in attention scores, making experiments rigorous and credible.

Weaknesses:
Incomplete description of hyperparameter settings. Although the authors have open-sourced the experiment code, reproducing the experiments still requires some hyperparameter settings. Detailed hyperparameter settings for each experiment are hoped to be provided by the authors.

Some professional or targeted concepts ("bimodal trend", Kolmogorov-Smirnov, etc.) are insufficiently described, which may lead to difficulties in understanding for readers. It is hoped that explanations can be strengthened.

**Questions For Authors:**

In some of the experimental demonstrations (Figure 1, Figure 2, Figure 3, Figure 5), three datasets (roman-empire, amazon-ratings, wikics) were selected and showed different forms of performance. Could you comprehensively explain how these different datasets performed across multiple experiments?

Similar to the previous question, why were these specific datasets chosen? Were other types of datasets also tested but not shown in the paper?

The article repeatedly mentions attention scores between different tokens. Can you elaborate on how these scores are specifically calculated? Are they calculated as normal attention layer scores? Are there any special cases?

The appendix shows charts of attention scores for first-order nodes and their child nodes at different layers (Layer 1 and Layer 3). In deeper network layers, would this pattern of attention scores differ?

**Relation To Broader Scientific Literature:**

In the research of LLM attention mechanisms, the author has identified a phenomenon termed "Skewed Line Sink" within graph datasets, which hinders the attention mechanism's ability to focus on nodes effectively.
In the context of graph machine learning research, the author provides insightful analysis and offers multiple methods and metrics for evaluating LLMs designed for graph data.

**Theoretical Claims:**

Yes, I have checked the data metrics in Table 1, Table 2, and Table 3 of the article, and they are consistent with the hypotheses and claims proposed at the beginning of the paper.

---

> ### Author Rebuttal · Authors · 2025-03-31
>
> >In some of the experimental demonstrations (Figure 1,2,3,5), three datasetswere selected and showed different forms of performance. Could you comprehensively explain how these different datasets performed across multiple experiments?
>
> We are happy to address your question. In our experimental demonstrations, the LLM exhibited strong performance on the Roman-Empire dataset (Table 2 - overcoming confounding factors, Figure 2 - enhancing attention scores for structural information). Additionally, in other visualization analyses, it demonstrated an understanding of graph structures. However, on other datasets (Pubmed, Wikics), the model showed disordered and difficult-to-understand behavior.
>
> > The article repeatedly mentions attention scores between different tokens. Can you elaborate on how these scores are specifically calculated? Are they calculated as normal attention layer scores? Are there any special cases?
>
> As mentioned in our previous response, we selected datasets that the LLM could handle well (Roman-Empire) and those it struggled with (Pubmed, Wikics). The goal was to identify differences between the two and explore how the LLM successfully understood graph structures.
>
>
> > The appendix shows charts of attention scores for first-order nodes and their child nodes at different layers (Layer 1 and Layer 3). In deeper network layers, would this pattern of attention scores differ?
>
> Indeed, during our experiments, we observed that the unique behaviors of deeper attention networks became more pronounced compared to shallower ones, such as the "attention sink" and "Skewed Line Sink."
>
> > Some professional or targeted concepts ("bimodal trend", Kolmogorov-Smirnov, etc.) are insufficiently described. It is hoped that explanations can be strengthened.
>
> Thank you for pointing this out. Insufficient explanation of certain professional terms and concepts may indeed cause difficulties for readers. Below are more detailed explanations of the mentioned concepts:
>
> Bimodal Trend: The appearance of a bimodal trend in the distribution of attention scores might indicate that the model pays particular attention to specific types of nodes or positions, while others form another cluster.
> Kolmogorov-Smirnov Test: This is a non-parametric test used to compare whether two samples come from the same distribution. The KS test is most commonly used to determine whether two sets of observations are significantly different or whether one set of observations differs significantly from a theoretical distribution.
>
> In addition, we have further revised Attention Window and JS.
>
> > Although the authors have open-sourced the experiment code, reproducing the experiments still requires some hyperparameter settings. Detailed hyperparameter settings for each experiment are hoped to be provided by the authors.
>
> Thank you for your thorough consideration. Due to the length constraints of the rebuttal, we have made corrections in the revised version.

---

### Official Review · Reviewer_tVm1 · 2025-03-14

**Overall Recommendation:** 3

**Summary:**

The paper empirically studies the behavior of the attention mechanism on graph inputs. First, they show that the attention distribution changes after finetuning, therefore concluding that LLMs can recognize graphs. Second, they show that altering the connectivity does not affect significantly the performance, showcasing that LLMs tend to not utilize the connectivity. Third, they uncovers that the attention does not match the graph structure. Finally, they introduce the concept of global linkage horizon, which indicates the visibility range of node tokens under the attention window, and study the impact of different values.




## update after rebuttal

Increased my score to weak accept after rebuttal, since the authors experimented with different prompts as asked.

**Claims And Evidence:**

The claims are supported by evidence.

**Essential References Not Discussed:**

None

**Experimental Designs Or Analyses:**

The experimental designs are valid.

**Methods And Evaluation Criteria:**

The proposed evaluations make sense. However, in my opinion the paper overlooks the impact of the prompt design, which is responsible of ensuring that graph-structured data can be presented in a way that can be consumed by LLMs, and that has been shown to have significant impact on the performance [1].


[1] Fatemi et al., 2023 Talk like a Graph: Encoding Graphs for Large Language Models

**Other Comments Or Suggestions:**

1. Please change the running title in the submission, as it currently is Submission and Formatting Instructions for ICML 2025.
2. I think you should replace training with finetuning in Q1, as if I understand correctly you are not training the LLM from scratch.
3. Aattention -> Attention in the captions of Figures 7 to 12.

**Other Strengths And Weaknesses:**

The clarify of the paper can be significantly improved. For instance, what first-order and second-order nodes are is never introduced. From the context, sometimes it refers to first-order and second-order neighbors, while other times it is unclear.

**Questions For Authors:**

Please discuss the prompts used and their impact on your findings. I see that Appendix J presents the prompt, but does not analyze the impact of changing it, especially because the assistant part can be significantly modified.

**Relation To Broader Scientific Literature:**

Studying the behavior of LLMs on graph data is an interesting topic, which has been recently studied in numerous works. However, I think the findings should be conditioned on the particular prompt used.

**Theoretical Claims:**

There are no theoretical claims.

---

> ### Author Rebuttal · Authors · 2025-03-31
>
> > Please discuss the prompts used and their impact on your findings. I see that Appendix J presents the prompt, but does not analyze the impact of changing it, especially because the assistant part can be significantly modified.
>
> Thank you for your suggestions regarding the experimental section of our paper. We **designed various Prompt formats for analysis and discussion, inspired by the recommended paper (TALK LIKE A GRAPH)**, and supplemented these into our experiments.
>
> (1) Our standard prompt (Appendix J Prompt)
> (2) Adjacency. Using integer node encoding and parenthesis edge encoding.
> (3) Expert. Employing alphabet letters for node encoding and arrows as edge encoding. The encoding starts with “You are a graph analyst”
> (4) Incident. Using integer node encoding and incident edge encoding.
>
> We conducted analyses for Table 2 (disruption experiment) and Figure 3 (attention distribution) from the paper.
>
> The results of the disruption experiment (Table 2) are as follows. From the results, it can be seen that although different templates have varying model performances, **the overall trend is consistent**, showing utilization of link information specifically on the Roman dataset, while link information did not play a role on most other datasets. (PS: Our initial template (1) remains the best performing and makes the best use of link information.)
>
>
>
> | Dataset       | Raw        | (I)        | (II)       | (III)      | (IV)       |
> |---------------|------------|------------|------------|------------|------------|
> **(1)**
> | Wikics        | 0.7862 ± 0.007 | 0.7847 ± 0.004 | 0.7907 ± 0.0035| 0.7670 ± 0.003 | 0.7087 ± 0.005|
> | Pubmed        | 0.8367 ± 0.003 | 0.8363 ± 0.001 | 0.8364 ± 0.002 | 0.7698 ± 0.003 | 0.7835 ± 0.001|
> | Amazon-Ratings| 0.4486 ± 0.002 | 0.3980 ± 0.003 | 0.3977 ± 0.002 | 0.3915 ± 0.003 | 0.3813 ± 0.001|
> | Roman-Empire  | 0.8089 ± 0.001 | 0.7918 ± 0.001 | 0.7910 ± 0.002 | 0.6290 ± 0.002 | 0.5784 ± 0.002|
> **(2)**
> | Wikics        | 0.7795 ± 0.006 | 0.7789 ± 0.005 | 0.7778 ± 0.003 | 0.7512 ± 0.004 | 0.6871 ± 0.006|
> | Pubmed        | 0.8112 ± 0.004 | 0.8108 ± 0.002 | 0.8050 ± 0.001 | 0.7745 ± 0.002 | 0.7783 ± 0.002|
> | Amazon-Ratings| 0.4231 ± 0.003 | 0.4038 ± 0.002 | 0.3923 ± 0.003 | 0.3869 ± 0.002 | 0.3768 ± 0.002|
> | Roman-Empire  | 0.7932 ± 0.002 | 0.7961 ± 0.002 | 0.7935 ± 0.001 | 0.7228 ± 0.003 | 0.6130 ± 0.001|
> **(3)**
> | Wikics        | 0.7806 ± 0.007 | 0.7793 ± 0.004 | 0.7851 ± 0.0035| 0.7618 ± 0.003 | 0.7025 ± 0.005|
> | Pubmed        | 0.8315 ± 0.003 | 0.8310 ± 0.001 | 0.8312 ± 0.002 | 0.7643 ± 0.003 | 0.7781 ± 0.001|
> | Amazon-Ratings| 0.4438 ± 0.002 | 0.3932 ± 0.003 | 0.3929 ± 0.002 | 0.3867 ± 0.003 | 0.3765 ± 0.001|
> | Roman-Empire  | 0.8030 ± 0.001 | 0.7864 ± 0.001 | 0.7856 ± 0.002 | 0.6234 ± 0.002 | 0.5728 ± 0.002|
> **(4)**
> | Wikics        | 0.7811 ± 0.006 | 0.7833 ± 0.005 | 0.7812 ± 0.0030| 0.7665 ± 0.004 | 0.7092 ± 0.006|
> | Pubmed        | 0.8325 ± 0.004 | 0.8322 ± 0.002 | 0.8317 ± 0.001 | 0.7897 ± 0.002 | 0.7064 ± 0.002|
> | Amazon-Ratings| 0.4379 ± 0.003 | 0.4085 ± 0.002 | 0.4072 ± 0.003 | 0.3978 ± 0.002 | 0.3885 ± 0.002|
> | Roman-Empire  | 0.8091 ± 0.002 | 0.7915 ± 0.002 | 0.7913 ± 0.001 | 0.6285 ± 0.003 | 0.5788 ± 0.001|
>
>
> The visualization for attention distribution (Figure 3) has been placed in an anonymous link
> [anonymous link](https://anonymous.4open.science/LLMexploration-B21F)
>
> **Additionally, it should be noted** that the input method we selected (Appendix J Prompt) was inspired by LLaGa (ICML), InstructGLM (EACL). **This is a recognized direct method within the research community to better assess the LLM's understanding of every node in the graph structure input.** It possesses **uniqueness in positional certainty**: it will not cause changes in LLM understanding due to different embedding methods; **graph structure position determinability**: the graph location of nodes can be determined through prompts, and **passes the WL-test**.
>
>
> > The clarify of the paper can be significantly improved. For instance, what first-order and second-order nodes are is never introduced. From the context, sometimes it refers to first-order and second-order neighbors, while other times it is unclear.
>
> Thank you for your revision suggestions. We have corrected the clarity throughout the paper and s**tandardized the naming of first-order nodes**.
>
>
> > Please change the running title in the submission, as it currently is Submission and Formatting Instructions for ICML 2025.
>
> Thank you for your advice. We have revised this accordingly.
>
>
> > I think you should replace training with finetuning in Q1, as if I understand correctly you are not training the LLM from scratch.
>
> We have revised the expressions throughout the paper to more accurately reflect fine-tuning, thanks for your suggestion.

---

> > ### Comment · Reviewer_tVm1 · 2025-04-02
> >
> > Can you therefore clarify what are first-order nodes?

---

> > > ### Author Response · Authors · 2025-04-03
> > >
> > > We will further explain the concepts of **"central node," "first-order nodes," and "second-order nodes"**:
> > >
> > > In graph-structured data, the **"central node"** (or "task node") refers to the core node that we focus on, which is typically **the target of analysis or a specific task**. For example, in a social network, if we are studying the influence of a particular user, that user is the central node.
> > >
> > > **"First-order nodes"** refer to the nodes **directly connected to the central node**. In other words, these are the neighboring nodes within one hop. For instance, in a social network graph, a user's direct friends are their first-order nodes. These first-order nodes usually have a direct impact on the central node and play an important role in many graph analysis tasks.
> > >
> > > **"Second-order nodes,"** on the other hand, refer to the nodes **directly connected to the first-order nodes** but **not directly connected to the central node**. In other words, second-order nodes are those two hops away from the central node. Taking the social network as an example, if you consider your friends as first-order nodes, then your friends' friends (excluding mutual friends) are second-order nodes. Although these second-order nodes do not directly connect to the central node, they indirectly influence the behavior or attributes of the central node through the first-order nodes.
> > >
> > > Finally, if you are satisfied with our response, could you please consider improving your evaluation of our work?

---

### Official Review · Reviewer_aPph · 2025-03-17

**Overall Recommendation:** 1

**Summary:**

This paper does an empirical investigation of attention patterns when an LLM is applied to graph problems.  They find that their treatment of graphs in LLMs is not very sensitive to graph connectivity information.  Some improvement is gained with attention masking.

**Claims And Evidence:**

One fundamental problem with this paper is that they make very general claims based on very specific experiments.  For example, they only consider one approach to inputting graphs to LLMs, and then claim that LLMs don't understand graphs very well.  Given the lack of detail about how they input graphs to the LLM, it is hard to even know what the narrower claim should be.

**Essential References Not Discussed:**

All the work on processing graphs in transformers.

**Experimental Designs Or Analyses:**

I will put my main weakness in this box, since it relates to the setup of the experiments.

The paper reads as if the authors simply forgot to include a section on the model.  Even after looking at the Appendix, when I started reading the analysis (Section 3A) I couldn't figure out how they input the graph connectivity to the LLM.  Without an understanding of the model they are evaluating, it is impossible to understand the relevance of the results.
In the middle of the analysis (Section 3B), Figure 4 seems to give some hint about how this is done.  Given the results, I conclude that this way of inputting the graph connectivity simply doesn't work very well.  Given what we already know about how transformers process information, this is not surprising.

**Methods And Evaluation Criteria:**

The methods are various ways to characterize the attention maps, which don't seem inappropriate in and of themselves.

**Other Comments Or Suggestions:**

I don't see any U-shaped curves in Figure 3.

**Other Strengths And Weaknesses:**

The main weaknesses have been discussed above, even if they are more general than the headings for those boxes.
Essentially, I don't feel I learned much from the analysis given in this paper because it does not sufficiently specify the model being analysed.

The literature review is not very scholarly.  Some citations are missing years.  Many don't appear to be reviewed publications.  Almost all are from the past 2 years.

**Questions For Authors:**

What does "node sampling configured as 8x8" mean?

**Relation To Broader Scientific Literature:**

There is a substantial literature of how to process graphs with a transformer architecture ("graph transformers", or "graph-to-graph transformers"), which the authors seem to be unaware of.  That work shows that transformers process graph relations using their attention functions, so graph connectivity can be input through their attention weight calculations.

This probably explains why they get better results with graph-based masked attention instead of full connectivity.  Masking attention weights is a method to input graph connectivity, which compensates for the inadequate method which they use otherwise.

**Theoretical Claims:**

NA

---

> ### Author Rebuttal · Authors · 2025-03-31
>
> Thank you for your valuable feedback. However, there seems to be some misunderstanding regarding our study's objectives. The content you claimed is missing is mentioned in the appendices. Below are our responses to clarify the points raised.
>
> >What does "node sampling configured as 8x8" mean?
>
> Node sampling is a common technique in graph machine learning used for efficient model training. To avoid high computational costs associated with training on an entire graph, node sampling selects a subset of nodes and their neighbors for training. Specifically, we sample 8 neighbors for a central node (forming first-order nodes) and then sample 8 more nodes for each of these first-order nodes (forming second-order nodes). This results in a total of 1 (central node) + 8 (first-order nodes) + 64 (second-order nodes). For LLMs processing graph data, an 8x8 **sampling configuration is widely considered reasonable** [1][2].
>
> >This paper makes very general claims based on very specific experiments. It only considers one approach to inputting graphs into LLMs and then claims that LLMs don't understand graphs well.
>
> The method we chose for **input (Appendix J Prompt) is based on LLaMA (ICML) [1] and InstructGLM (EACL) [2]**, which are recognized by the research community as straightforward approaches that can better evaluate an LLM's ability to understand each node in graph-structured inputs. This method has **unique positional properties**: it does not alter the LLM’s understanding due to different embedding methods, **offers graph structural position certainty** (the graph position of a node can determine its prompt position, and vice versa), and can **pass the WL-test**. Additionally, we **arrived at general conclusions after conducting extensive experiments**, including perturbation tests, shuffling experiments, attention window analyses, and various visualization studies, which align with common research paradigms. Meanwhile, we have added experimental comparisons of different input methods under the first response to **Reviewer pYJ8**.
>
> >The paper reads as if the authors forgot to include a section on the model. Even after looking at the Appendix, I couldn't figure out how they input the graph connectivity to the LLM.
>
> In lines 267-269, we reference LLaMA [1] and InstructGLM [2], foundational works in the LLM4Graph field that have **gained broad acceptance**. Detailed Prompt templates are provided in **Appendix J**, which is referenced in the "Other work in Appendix" section of the main text.
>
> >Many citations do not appear to be reviewed publications; almost all are from the past two years.
>
> LLM4Graph is a rapidly evolving field where many impactful papers first appear on Arxiv. All 10 Arxiv references we cited have been accepted to conferences such as **ICML, NIPS, and KDD**, awaiting publication. Only 2 papers remain under review. Given that LLMs were introduced to the GNN community around early 2023, most citations are recent.
>
> >There is substantial literature on processing graphs with transformer architectures ("graph transformers"), which the authors seem unaware of.
>
> It may be noted that we discuss the application of transformer architectures (not just LLMs) to graph data in Appendix C, referenced in the "Other work in Appendix" section of the main text.
>
> >Given the results, I conclude that this way of inputting graph connectivity simply doesn't work very well.
>
> We agree that simply input methods may not yield optimal performance, but our aim was to straightforwardly analyze underlying issues rather than achieve sota results. Our chosen method serves as a recognized baseline for further research.
>
> >I don't see any U-shaped curves in Figure 3.
>
> As explained in our paper, "attention towards node tokens exhibits a U-shaped distribution or slash trend." Lower attention scores for initial nodes are attributed to their position towards the end of the text, resulting in less pronounced U-shapes or trends leaning towards slashes, a common observation also noted in similar studies [3]. We maintained this representation without adjustments, as detailed in our text.
>
>
> [1] Chen R, Zhao T, JAISWAL A K, et al. LLaGA: Large Language and Graph Assistant[C]//Forty-first International Conference on Machine Learning.
> [2] Ye R, Zhang C, Wang R, et al. Language is All a Graph Needs[C]//18th Conference of the European Chapter of the Association for Computational Linguistics, EACL 2024-Findings of EACL 2024. Association for Computational Linguistics (ACL), 2024: 1955-1973.
> [3] Hsieh C Y, Chuang Y S, Li C L, et al. Found in the middle: Calibrating Positional Attention Bias Improves Long Context Utilization[C]//Findings of the Association for Computational Linguistics ACL 2024. 2024: 14982-14995.

---

### Decision · Program_Chairs · 2025-05-01

**Decision:**

Accept (poster)

**Comment:**

This paper empirically studies how LLMs process graph-structured data from the perspective of attention mechanisms.
Almost all reviewers believe that this is a good paper, so I recommend acceptance. I also recognize that there is a growing community, originating from the GNN field, that is interested in how LLMs perform on relational data.

That said, I agree with reviewer aPph that there are general claims being made in the paper, while only a particular way of encoding relational data as input is explored in the submitted version. If the paper is accepted, I would strongly recommend that the authors clearly state their input setting and remove the generic claims about failure, which are also present in the abstract.